# Aberrant upregulation of the glycolytic enzyme PFKFB3 in CLN7 neuronal ceroid lipofuscinosis

Irene Lopez-Fabuel [1,2,12✉], Marina Garcia-Macia [1,2,3,12], Costantina Buondelmonte[1,2], Olga Burmistrova[4], Nicolo Bonora[1,2], Paula Alonso-Batan[1,2], Brenda Morant-Ferrando[1,2], Carlos Vicente-Gutierrez [1,2,3], Daniel Jimenez-Blasco[1,2,3], Ruben Quintana-Cabrera [1,2,3], Emilio Fernandez[1,2,3], Jordi Llop[5], Pedro Ramos-Cabrer [5,6], Aseel Sharaireh[7], Marta Guevara-Ferrer [7], Lorna Fitzpatrick[7], Christopher D. Thompton[7], Tristan R. McKay[7], Stephan Storch[8], Diego L. Medina [9,10], Sara E. Mole [11], Peter O. Fedichev[4], Angeles Almeida [1,2] & Juan P. Bolaños [1,2,3✉]

CLN7 neuronal ceroid lipofuscinosis is an inherited lysosomal storage neurodegenerative disease highly prevalent in children. *CLN7/MFSD8* gene encodes a lysosomal membrane glycoprotein, but the biochemical processes affected by CLN7-loss of function are unexplored thus preventing development of potential treatments. Here, we found, in the *Cln7ᐞᵉˣ²* mouse model of CLN7 disease, that failure in autophagy causes accumulation of structurally and bioenergetically impaired neuronal mitochondria. In vivo genetic approach reveals elevated mitochondrial reactive oxygen species (mROS) in *Cln7ᐞᵉˣ²* neurons that mediates glycolytic enzyme PFKFB3 activation and contributes to CLN7 pathogenesis. Mechanistically, mROS sustains a signaling cascade leading to protein stabilization of PFKFB3, normally unstable in healthy neurons. Administration of the highly selective PFKFB3 inhibitor AZ67 in *Cln7ᐞᵉˣ²* mouse brain in vivo and in CLN7 patients-derived cells rectifies key disease hallmarks. Thus, aberrant upregulation of the glycolytic enzyme PFKFB3 in neurons may contribute to CLN7 pathogenesis and targeting PFKFB3 could alleviate this and other lysosomal storage diseases.

---

[1] Institute of Functional Biology and Genomics (IBFG), Universidad de Salamanca, CSIC, Salamanca, Spain. [2] Institute of Biomedical Research of Salamanca (IBSAL), Hospital Universitario de Salamanca, Salamanca, Spain. [3] Centro de Investigación Biomédica en Red de Fragilidad y Envejecimiento Saludable (CIBERFES), Madrid, Spain. [4] Gero Discovery LLC, Moscow, Russia. [5] CIC biomaGUNE, Basque Research and Technology Alliance (BRTA), Donostia-San Sebastián, Spain. [6] Ikerbasque, Basque Foundation for Science, Bilbao, Spain. [7] Centre for Bioscience, Manchester Metropolitan University, Manchester M1 5GD, UK. [8] University Children's Research@Kinder-UKE, University Medical Center Hamburg-Eppendorf, Hamburg, Germany. [9] Telethon Institute of Genetics and Medicine (TIGEM), High Content Screening Facility, Via Campi Flegrei 34, 80078 Pozzuoli, Italy. [10] Medical Genetics Unit, Department of Medical and Translational Science, Federico II University, 80138 Naples, Italy. [11] MRC Laboratory for Molecular Biology and GOS Institute of Child Health, University College London, London, UK. [12] These authors contributed equally: Irene Lopez-Fabuel, Marina Garcia-Macia. ✉email: ilofa@usal.es; jbolanos@usal.es

The neuronal ceroid lipofuscinoses (NCLs) are a family of monogenic life-limiting pediatric neurodegenerative disorders collectively known as Batten disease[1]. Although genetically heterogeneous[2], NCLs share several clinical symptoms and pathological hallmarks such as lysosomal accumulation of lipofuscin and astrogliosis[2,3]. Ceroid lipofuscinosis, neuronal 7 (CLN7) disease belongs to a group of NCLs that present in late infancy[4–6] and, whereas *CLN7/major facilitator superfamily domain containing 8 (MFSD8)* gene is known to encode a lysosomal membrane glycoprotein[4,7–9], the biochemical processes affected by CLN7-loss of function are unexplored, which has hampered the development of therapeutic interventions[1,10]. Forty-six disease-causing mutations are recorded in the NCL mutation database (ucl.ac.uk/ncl-disease) in *CLN7/MFSD8*, causing a broad phenotypic range, from classic late infantile CLN7 disease to non-syndromic retinal disease with onset in childhood or as late as the 7th decade[4]. Given that treatment for CLN7 disease is likely to be more challenging than for NCLs encoding lysosomal enzymes such as ceroid lipofuscinosis, neuronal 2 (CLN2)/tripeptidyl peptidase 1 (TPP1)[11] here we aimed to understand the biochemical processes affected in CLN7 disease. Here, using the *Cln7^Δex2* mouse model of CLN7 disease, we found an aberrant upregulation of pro-glycolytic enzyme PFKFB3 in neurons that may contribute to CLN7 pathogenesis.

## Results

### Failure in autophagy causes accumulation of structural and functionally impaired mitochondria in *Cln7^Δex2* mouse.

In Cln7-null neurons in primary culture from *Cln7^Δex2* mice[12] (Supplementary Fig. 1a), the mitochondrial indicators ATP synthase-subunit c (SCMAS) and heat-shock protein-60 (HSP60) co-localized with the lysosome-associated membrane protein 1 (LAMP1) (Fig. 1a and Supplementary Fig. 1b), suggesting lysosomal accumulation of mitochondria. Inhibition of lysosomal proteolysis increased the protein levels of the autophagosome marker LC3-II in wild-type (WT), but not in *Cln7^Δex2* neurons (Fig. 1b and Supplementary Fig. 1c), indicating an impairment in the macro-autophagy (hereafter, autophagy) previously observed in lysosomal-storage disorders[13,14]. To assess whether this failure in autophagy affected mitochondrial turnover, SCMAS and HSP60 abundances were determined in neurons incubated with the lysosomal inhibitors. As shown in Fig. 1c and Supplementary Fig. 1d, lysosomal inhibition triggered the accumulation of SCMAS and HSP60 in WT neurons, indicating mitophagy flux[15]. However, these mitochondrial markers were already increased in untreated *Cln7^Δex2* neurons and were little affected by inhibiting lysosomal function (Fig. 1c and Supplementary Fig. 1d). In addition, PTEN-induced kinase-1 (PINK1) 63/53 ratio[16] and Parkin[17] increased in *Cln7^Δex2* neuronal mitochondria (Supplementary Fig. 1e). These data suggest that the mitochondrial clearance in *Cln7^Δex2* neurons is impaired. The metabolic profile analysis revealed a decrease in the basal oxygen consumption rate (OCR), ATP-linked and maximal OCR, and proton leak in *Cln7^Δex2* neurons (Fig. 1d), indicating bioenergetically impaired mitochondria. The specific activities of the mitochondrial respiratory chain (MRC) complexes (Supplementary Fig. 1f) were unchanged in the *Cln7^Δex2* neurons. However, isolation of mitochondria followed by blue-native gel electrophoresis (BNGE), complex I (CI) in-gel activity assay (IGA), and western blotting, revealed CI disassembly from mitochondrial super-complexes (SCs) in *Cln7^Δex2* neurons (Fig. 1e). These data confirm the decreased mitochondrial energy efficiency[18] and suggest the increased formation of mitochondrial reactive oxygen species (mROS)[19] in *Cln7^Δex2* neurons. Flow cytometric analysis of mROS (Fig. 1f; see also Supplementary Fig. 1g for unchanged mitochondrial membrane potential) and fluorescence analysis of $H_2O_2$

(Supplementary Fig. 1h), confirmed mROS enhancement in *Cln7^Δex2* neurons. Given the cross-talk between ROS and endoplasmic reticulum (ER) stress in disease[20], we investigated whether *Cln7^Δex2* neurons suffered from ER stress. Real-time-quantitative polymerase chain reaction (RT–qPCR) analysis of the unfolded protein response (UPR), which accumulate during ER stress[21], showed no changes in *Cln7^Δex2* neurons (Supplementary Fig. 1i). Given that cultured neurons do not necessarily behave exactly as they do in vivo, we validated our observations in the *Cln7^Δex2* mouse model in vivo. Thus, to characterize mitochondria from *Cln7^Δex2* mice in vivo, we next performed electron microscopy analyses of the brain cortex (Supplementary Fig. 1j) before and after the onset of the immunohistochemical and behavioral symptoms of the disease[12]. We found larger and longer brain mitochondria in the pre-symptomatic *Cln7^Δex2* mice, an effect that proceeded with age (Fig. 1g and Supplementary Fig. 1k), suggesting progressive mitochondrial swelling. CI disassembly from SCs in brain mitochondria (Fig. 1h) and increased mROS in freshly purified neurons from the adult brain (Fig. 1i and Supplementary Fig. 1l) were confirmed in *Cln7^Δex2* mice. Altogether, these findings suggest that Cln7 loss causes impaired autophagic clearance of brain mitochondria leading to the aberrant accumulation of structurally disorganized, bioenergetically impaired, and high ROS-generating organelle.

### Increased generation of mitochondrial ROS by neurons accounts for impaired mitochondrial accumulation and hallmarks of CLN7 disease in *Cln7^Δex2* mouse in vivo.

Next, we assessed the impact of excess neuronal mROS on CLN7 disease progression. *Cln7^Δex2* mice were thus crossed with mice expressing a mitochondrial-tagged isoform of the $H_2O_2$-detoxifying enzyme catalase (mCAT) governed by the neuron-specific[22] calcium/calmodulin-dependent protein kinase II alpha (CaMKIIa) promoter (*CaMKIIa^Cre-mCAT^LoxP*). mCAT efficacy in vivo was previously validated[23]. The resulting progeny (*Cln7^Δex2-CAMKIIa^Cre-mCAT*) was analyzed and compared with littermate *Cln7^Δex2-mCAT^LoxP* and control (*mCAT^LoxP* and *CAMKIIa^Cre-mCAT*) mice. The increased mROS observed in *Cln7^Δex2-mCAT^LoxP* neurons was abolished in *Cln7^Δex2-CAMKIIa^Cre-mCAT* neurons (Fig. 2a and Supplementary Fig. 2a), verifying the efficacy of this approach. Brain mitochondrial swelling was confirmed in *Cln7^Δex2-mCAT^LoxP* mice (Fig. 2b), which also showed mitochondrial cristae profile widening (Fig. 2b), a phenomenon previously observed in cells with bioenergetically-inefficient mitochondria[24,25]. Both the mitochondrial swelling and cristae profile widening observed in the *Cln7^Δex2-mCAT^LoxP* mice were rescued by expressing mCAT in neurons of the *Cln7^Δex2-CAMKIIa^Cre-mCAT* mice (Fig. 2b). Thus, neuronal mROS participates in the accumulation of functional and ultrastructural impaired mitochondria in *Cln7^Δex2* mouse brain. In line with this, the increase in SCMAS abundance observed in the *Cln7^Δex2-mCAT^LoxP* mouse brain (Fig. 2c and Supplementary Fig. 2b) was abolished, or partially restored, in *Cln7^Δex2-CAMKIIa^Cre-mCAT* mice (Fig. 2c and Supplementary Fig. 2b). Brain SCMAS accumulation in autofluorescent ceroid lipopigments (lipofuscin)-containing lysosomes is a hallmark of Batten disease[2] with some exemptions[26]. Consistently with this notion, lipofuscin was accumulated in the brain of *Cln7^Δex2-mCAT^LoxP* mice, an effect that was ameliorated in *Cln7^Δex2-CAMKIIa^Cre-mCAT* mice (Fig. 2c and Supplementary Fig. 2c). Moreover, activation of astrocytes and microglia is another hallmark of Batten disease[3] that is mimicked in the brain of *Cln7^Δex2* mice[12]. We found increased glial-fibrillary acidic protein (GFAP) and ionized calcium-binding adaptor molecule-1 (IBA-1) proteins in the brain of *Cln7^Δex2-mCAT^LoxP* mice, suggesting astrocytosis and microgliosis, respectively; these effects were attenuated in *Cln7^Δex2-CAMKIIa^Cre-mCAT* mice (Fig. 2c and Supplementary

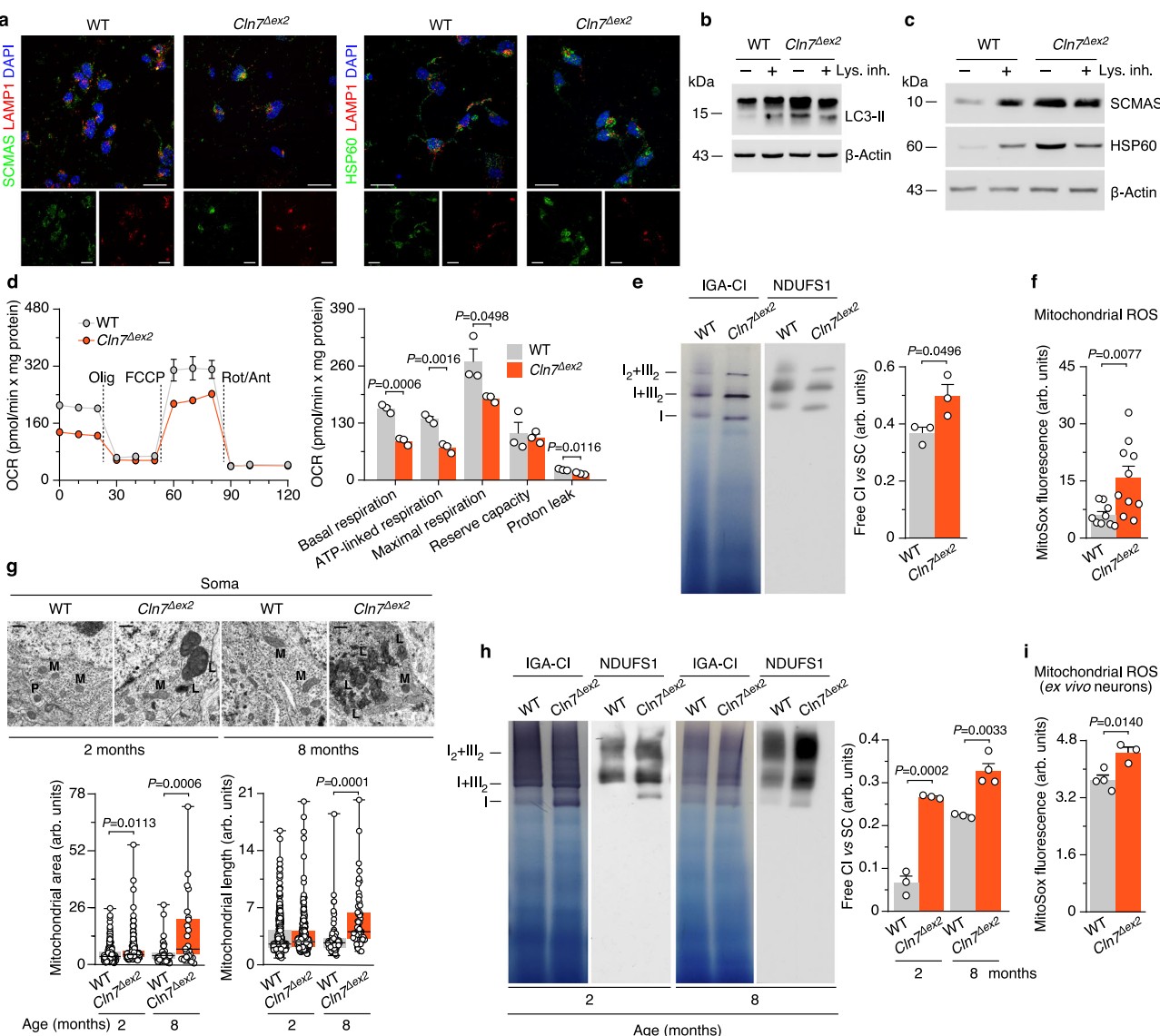

**Fig. 1 Failure in autophagy causes accumulation of structural and functionally impaired mitochondria in *Cln7^{Δex2}* mouse. a** SCMAS/LAMP1 and HSP60/SCMAS colocalization confocal analyses in primary neurons. DAPI reveals nuclei. Scale bar, 20 μm. **b** LC3-II western blot analysis in primary neurons incubated with lysosomal inhibitors leupeptin (100 μM) plus $NH_4Cl$ (20 mM) (Lys. Inh.) for 1 h (ß-actin, loading control). **c** HSP60 and SCMAS western blot analysis in primary neurons incubated with lysosomal inhibitors leupeptin (100 μM) plus $NH_4Cl$ (20 mM) (Lys. Inh.) for 1 h (ß-actin, loading control). **d** OCR analysis (left) and calculated parameters (right) in primary neurons. Data are mean ± SEM from $n = 3$ independent experiments. **e** Free complex I (CI) and CI-containing supercomplexes (SC) analyses in primary neurons by BNGE in-gel activity (IGA-CI) and by immunoblotted PVDF membranes against CI subunit NDUFS1. Data are mean ± SEM from $n = 3$ independent experiments. **f** Mitochondrial ROS analysis in primary neurons. Data are mean ± SEM from $n = 9$ (WT), $n = 10$ (*Cln7^{Δex2}*) independent experiments. **g** Representative electron microscopy images and analyses of mouse brain cortex mitochondria. Data are in box plots (the box extends from the 25th to 75th percentiles, the horizontal line indicates the median, and the whiskers go down to the smallest value and up to the largest) from $n \geq 27$ mitochondria per condition. Scale bar, 500 nm. (M mitochondria, L lysosome, P peroxisomes). **h** Free CI and CI-containing SC analyses of mouse brain cortex by BNGE IGA-CI and by immunoblotted PVDF membranes against NDUFS1. Data are mean ± SEM from $n = 3$ or $n = 4$ (*Cln7^{Δex2}*) 8-month old animals. **i** Mitochondrial ROS analysis in freshly isolated mouse brain cortex neurons. Data are mean ± SEM from $n = 3$ or $n = 4$ (WT) animals of 6-month old. Statistical analyses were performed by two-tailed Student's t test. Representative images and western blots out of $n \geq 3$ experiments are shown. See also Supplementary Fig. 1. Source data are provided as a Source Data file.

Fig. 2d). Altogether, these findings indicate that the generation of ROS by bioenergetically impaired mitochondria in *Cln7^{Δex2}* neurons contributes to the histopathological symptoms of CLN7 disease.

**Upregulation of PFKFB3 protein and activity via a $Ca^{2+}$/calpain/Cdk5 pathway sustains a high glycolytic flux in primary neurons obtained from *Cln7^{Δex2}* mice.** Mitochondrial ROS

stimulate brain glucose consumption through the glycolytic pathway in mouse[23]. In *Cln7^{Δex2}-mCAT^{LoxP}* neurons, both glycolysis (Fig. 3a), and its end-product lactate (Fig. 3b) were upregulated (by ~1.34 and ~1.64-fold, respectively), effects that were abolished in *Cln7^{Δex2}-CAMKIIa^{Cre}-mCAT* neurons (Fig. 3a, b). Glycolytic and pentose-phosphate pathway (PPP) fluxes are inversely regulated in neurons[27–29]. Agreeingly, the increased glycolytic flux observed in primary neurons obtained from *Cln7^{Δex2}* mice was accompanied by reduced PPP flux to a similar

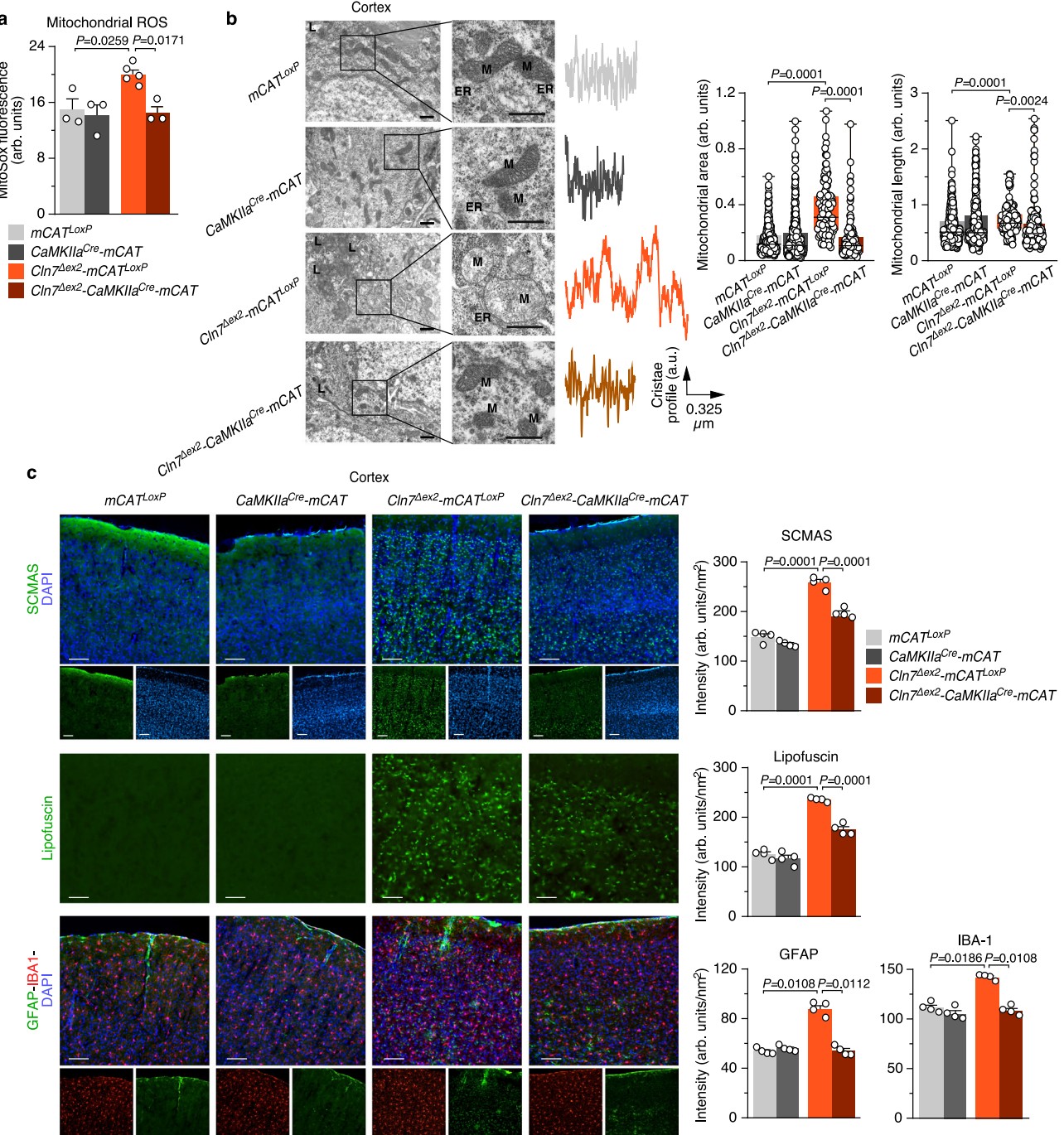

**Fig. 2 Increased generation of mitochondrial ROS by neurons accounts for impaired mitochondrial accumulation and hallmarks of CLN7 disease in *Cln7^Δex2* mouse in vivo. a** Mitochondrial ROS analysis in primary neurons from the designed genotype. Data are mean ± SEM from $n = 3$ or $n = 5$ (*Cln7^Δex2*-mCAT^LoxP) independent experiments. **b** Representative electron microscopy images of the mouse brain cortex displaying the cristae profile plot of intensities over the maximal axis of the magnified shown mitochondrion (left) and the analyses of mitochondrial area and length (right). Data are in box plots (the box extends from the 25th to 75th percentiles, the horizontal line indicates the median, and the whiskers go down to the smallest value and up to the largest) from $n \geq 136$ mitochondria per condition of 3-month-old mice. Scale bars, 600 nm. (M mitochondria, L lysosome, ER endoplasmic reticulum). **c** Representative images of SCMAS, lipofuscin, GFAP and IBA-1 immunohistochemical analysis of the mouse brain cortex. Data are mean ± SEM from $n = 4$ animals of 3-month old (three serial slices per mouse). Scale bar, 100 μm. Statistical analyses were performed by one-way ANOVA followed by Tukey's post hoc test. See also Supplementary Fig. 2. Source data are provided as a Source Data file.

extent (Fig. 3c). In vivo 2-[^18F]fluoro-2-deoxy-D-glucose ([^18F]FDG) uptake was unchanged in all analyzed brain areas of the *Cln7^Δex2* mouse, according to positron-emission tomography (PET) assessment (Supplementary Fig. 3a). However, in vivo ^1H-magnetic resonance spectroscopy ([^1H]MRS) analysis of the

*Cln7^Δex2* mouse brain revealed a twofold increase in the concentration of glycine (Supplementary Fig. 3b, c). Whilst the biosynthesis of glycine via the phosphorylated pathway requires glycolysis[30], its concentration is not a direct evidence of the glycolytic flux. Therefore, using [^18F]FDG-PET and [^1H]MRS,

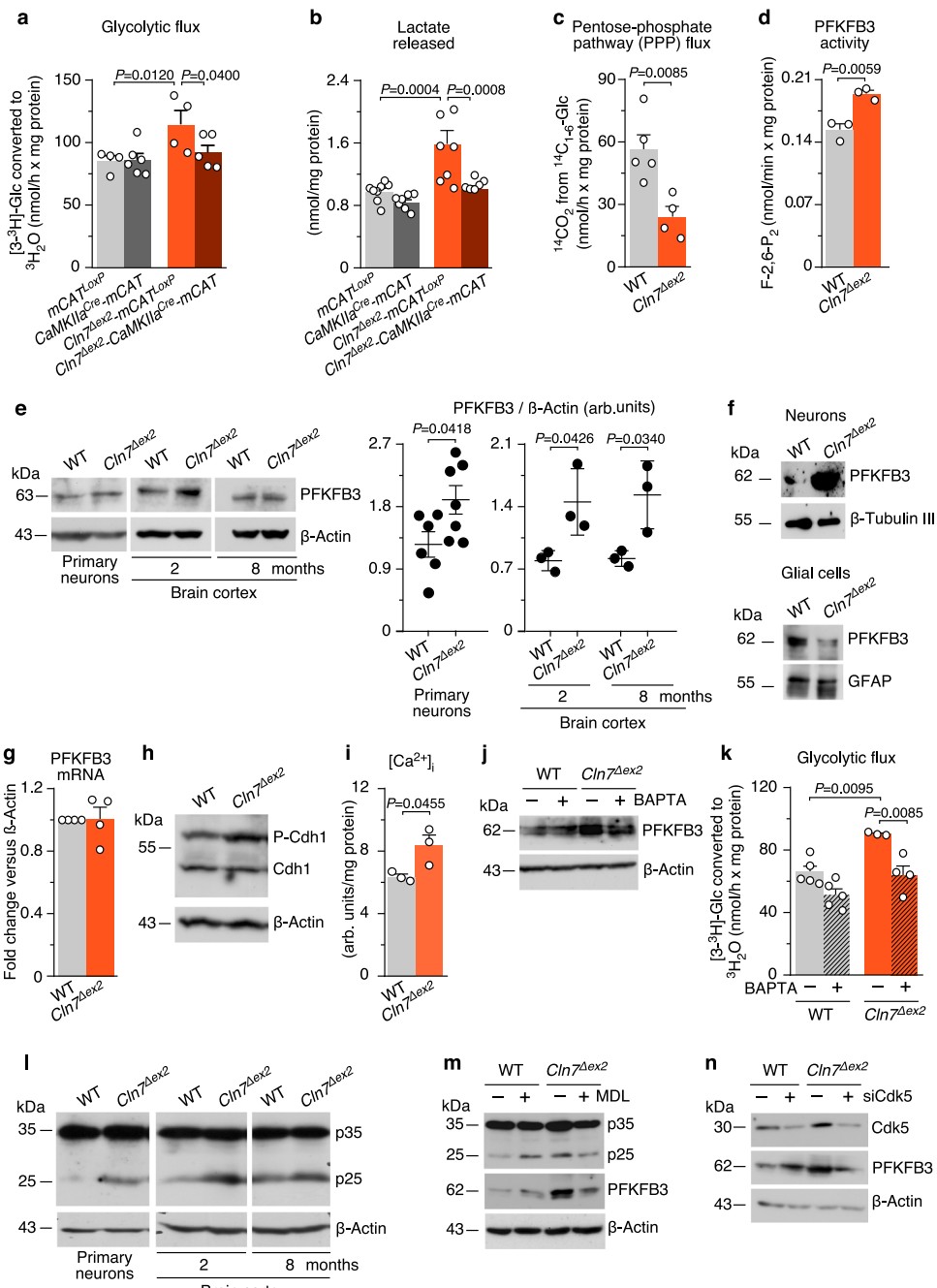

being approaches that lack cell-level resolution, failed to unambiguously ascertain in vivo upregulation of neuronal glycolysis in $Cln7^{\Delta ex2}$ mice. The increased glycolytic flux observed in primary neurons obtained from $Cln7^{\Delta ex2}$ mice can be indicative of hyperactive 6-phosphofructo-1-kinase (PFK1)[31,32], a rate-limiting step of glycolysis that is regulated by fructose-2,6-bisphosphate (F-2,6-$P_2$), a robust positive effector of PFK1[33]. The rate of F-2,6-$P_2$ formation was enhanced by ~1.27-fold in $Cln7^{\Delta ex2}$ neurons (Fig. 3d), a result that is compatible with higher activity of 6-phosphofructo-2-kinase/fructose-2,6-bisphosphatase-3 (PFKFB3) —i.e., the only F-2,6-$P_2$-forming isoenzyme found in neurons upon stress conditions[27]. PFKFB3 protein was increased both in primary neurons (~1.51-fold) and in vivo brain cortex (~1.85-fold) and cerebellum (~1.41-fold) (Fig. 3e and Supplementary Fig. 3d) of the $Cln7^{\Delta ex2}$ mice. To elucidate whether in vivo

PFKFB3 brain accumulation has neuronal or glial origin, we acutely separated these cell-type groups from the mouse cerebellum using an immunomagnetic approach. As shown in Fig. 3f and Supplementary Fig. 3e, PFKFB3 protein was found enhanced by ~4.77-fold in neurons and unaffected in the glia. Since PFKFB3 mRNA abundance was unaltered in $Cln7^{\Delta ex2}$ neurons (Fig. 3g), we conjectured that increased PFKFB3 protein could be the consequence of inactivating its degrading pathway[27]. $Cln7^{\Delta ex2}$ neurons showed hyperphosphorylation of the anaphase-promoting complex/cyclosome (APC/C) activator protein, Cdh1 (Fig. 3h and Supplementary Fig. 3f), which is sufficient to inhibit APC/C E3-ligase activity that targets PFKFB3 for proteasomal degradation[27]. To pursue this possibility, we noted that the $Ca^{2+}$-buffering capacity of bioenergetically compromised mitochondria is impaired[34]. Indeed, $Cln7^{\Delta ex2}$ neurons showed an enhanced

**Fig. 3 Upregulation of PFKFB3 protein and activity via a $Ca^{2+}$/calpain/Cdk5 pathway sustains a high glycolytic flux in $Cln7^{\Delta ex2}$ neurons. a** Glycolytic flux in primary neurons. Data are mean ± SEM from $n = 4$ ($mCAT^{LoxP}$, $Cln7^{\Delta ex2}$-$mCAT^{LoxP}$), $n = 6$ ($CaMKIIa^{Cre}$-mCAT) or $n = 5$ ($Cln7^{\Delta ex2}$-$CaMKIIa^{Cre}$-mCAT) independent experiments. **b** Lactate released by primary neurons ($n = 7$-8). Data are mean ± SEM from $n = 7$ ($CaMKIIa^{Cre}$-mCAT) or $n = 8$ independent experiments. **c** PPP flux in primary neurons. Data are mean ± SEM from $n = 5$ (WT) or $n = 4$ ($Cln7^{\Delta ex2}$) independent experiments. **d** Rate of P-2,6-$P_2$ formation in primary neurons. Data are mean ± SEM from $n = 3$ independent experiments. **e** Representative PFKFB3 western blot analysis in primary neurons and brain cortex (ß-actin, loading control) and the densitometric quantification of the bands (including the replicas). Data are mean ± SD from $n = 6$ (WT), $n = 7$ ($Cln7^{\Delta ex2}$) independent experiments, or $n = 3$ animals. **f** Representative western blots showing PFKFB3 protein abundances in immunomagnetically isolated neurons or glial cells (ß-tubulin III and glial-fibrillary acidic protein or GFAP, loading control for neurons and astrocytes, respectively). **g** PFKFB3 mRNA analysis by RT–qPCR in primary neurons. Data are mean ± SEM from $n = 4$ independent experiments (values normalized versus ß-actin). **h** Representative Cdh1 western blot analysis after PhosTag acrylamide electrophoresis in primary neurons (P-Cdh1, hyperphosphorylated Cdh1; ß-actin, loading control). **i** Cytosolic $Ca^{2+}$ analysis in primary neurons. Data are mean ± SEM from $n = 3$ independent experiments.
**j**, **k** Representative PFKFB3 western blot (**j**) and glycolytic flux (**k**) analyses in primary neurons incubated with $Ca^{2+}$ quelator BAPTA (10 μM; 1 h) (ß-actin, loading control). Data are mean ± SEM from $n = 5$ (WT), $n = 4$ ($Cln7^{\Delta ex2}$) independent experiments. **l** Representative p35 western blot revealing p35 and its cleavage product p25 in primary neurons and brain cortex (ß-actin, loading control). **m** Representative p35 and PFKFB3 western blot analyses in primary neurons incubated with calpain inhibitor MDL-28170 (MDL) (100 μM; 24 h) (ß-actin, loading control). **n** Representative Cdk5 and PFKFB3 western blot analyses in primary neurons transfected with Cdk5 siRNA (siCdk5) or scrambled siRNA (–) (9 nM; 3 days) (ß-actin, loading control). Statistical analyses performed by one-way ANOVA followed by DMS's (**a**) or Tukey's (**b**, **k**) post hoc tests or two-tailed Student's $t$ test (**c**, **d**, **e**, **g**, **i**). See also Supplementary Fig. 3. Source data are provided as a Source Data file.

concentration of cytosolic $Ca^{2+}$ (Fig. 3i), an activator of calpain—a proteolytic enzyme essential in the signaling cascade leading to Cdh1 hyperphosphorylation[35]. $Ca^{2+}$ sequestration reduced both PFKFB3 protein (Fig. 3j and Supplementary Fig. 3g) and glycolysis (Fig. 3k) in $Cln7^{\Delta ex2}$ neurons, confirming $Ca^{2+}$ involvement in increasing glycolytic flux. $Ca^{2+}$-mediated calpain activation proteolytically cleaves p35 into p25—a cofactor of the cyclin-dependent kinase-5 (Cdk5)[36] that phosphorylates Cdh1[35]. We found an increased p35 cleavage into p25 in $Cln7^{\Delta ex2}$ primary neurons and in vivo brain cortex and cerebellum (Fig. 3l and Supplementary Fig. 3h,i). Inhibition of calpain using the specific inhibitor[36] MDL-28170 rescued p35 cleavage and PFKFB3 increase (Fig. 3m and Supplementary Fig. 3j). Given that these effects suggest the involvement of Cdk5, Cdk5 was knocked down in $Cln7^{\Delta ex2}$ neurons, an action that prevented PFKFB3 increase (Fig. 3n and Supplementary Fig. 3k). Together, these results indicate the occurrence of a $Ca^{2+}$/calpain-mediated activation of Cdk5/p25 pathway that phosphorylates APC/C-cofactor Cdh1, eventually leading to the stabilization of glycolytic enzyme PFKFB3 in CLN7 disease.

**Pharmacological inhibition of PFKFB3 restores mitochondrial alterations and hallmarks of $Cln7^{\Delta ex2}$ disease in vivo.** In neurons, PFKFB3 destabilization boosts glucose consumption through PPP[27] and prevents damage-associated redox stress[27,37,38] given its role at supplying NADPH($H^+$)—an essential cofactor of glutathione regeneration[39,40]. We therefore sought to assess whether PFKFB3 activity is related to CLN7 disease. We undertook this by inhibiting PFKFB3 activity using the highly selective, rationally designed[41] compound AZ67. Incubation of $Cln7^{\Delta ex2}$ neurons with AZ67 at a concentration that inhibits PFKFB3 activity without compromising survival[42], prevented the increase in F-2,6-$P_2$ (Fig. 4a) and glycolysis (Fig. 4b) without affecting mROS (Fig. 4c and Supplementary Fig. 4a). Interestingly, AZ67 protected $Cln7^{\Delta ex2}$ neurons from activation of pro-apoptotic caspase-3 (Fig. 4d and Supplementary Fig. 4b), suggesting its potential therapeutic benefit. To test this in vivo, AZ67 was intracerebroventricularly administered in $Cln7^{\Delta ex2}$ mice daily for 2 months at a dose previously selected according to pharmacokinetic and safety parameters (Supplementary Fig. 4c, d, e). Electron microscopy analysis revealed that AZ67 did not affect the length or area of brain mitochondria in $Cln7^{\Delta ex2}$ mice (Fig. 4e), but it prevented the cristae profile amplitude reduction (Fig. 4e); this may indicate, as observed in other paradigms[24,25],

adaptation of the mitochondrial ultrastructure to a bioenergetically efficient configuration upon glycolysis inhibition. Incubation of $Cln7^{\Delta ex2}$ neurons with AZ67 partially restored the impairment in basal respiration (Fig. 4f), indicating a functional improvement of the mitochondria. In vivo, AZ67 prevented the accumulation of SCMAS, lipofuscin, and reactive astroglia in the cortex (Fig. 5a, b), and SCMAS and lipofuscin in the hippocampus and cerebellum (Supplementary Fig. 5a–c) of the $Cln7^{\Delta ex2}$ mice. Hindlimb paralysis[12] in $Cln7^{\Delta ex2}$ mice was prevented by AZ67 (Supplementary Movies 1–4), indicating functional recovery. Finally, to assess the possible translational implications of these results, neural precursors cells (NPCs) generated from induced pluripotent cells (iPCs) derived from control and two CLN7 disease patients homozygous for missense mutations (Fig. 6a–d) were analyzed. CLN7 patients-derived NPCs showed increased SCMAS staining (Fig. 6e) and mROS (Fig. 6f). Furthermore, these cells exhibited condensation of mitochondria in the perinuclear region (Fig. 6g), an effect that was rectified by AZ67 (Fig. 6h).

## Discussion

Here, we found that impaired autophagy pathway in CLN7 disease causes accumulation of dysfunctional mitochondria. These mitochondria exhibit complex I disassembly from supercomplexes, which accounts[19] for the high mROS production that contributes to CLN7 disease pathogenesis, according to the accumulation of SCMAS, lipofuscin, and astrogliosis. The signaling cascade involves a $Ca^{2+}$-mediated, calpain-promoted p25 formation, from p35 cleavage, that activated Cdk5. Active Cdk5 phosphorylates -and inactivates[35]- the E3-ligase APC/C-cofactor Cdh1, which leads[27] to PFKFB3 protein stabilization. Interestingly, Cdk5 also phosphorylates –and inhibits– collapsin response mediator protein 2 (CRMP2)[43], a cytoskeletal protein found reduced in the $CLN6^{nclf}$ mutant mouse model of Batten disease[44,45], suggesting a possible common mechanism in both NCLs. Increased PFKFB3 protein and activity stimulated, in primary neurons obtained from $Cln7^{\Delta ex2}$ mice, the flux of glycolysis, a pathway that in healthy neurons is downmodulated to facilitate glucose consumption through the antioxidant PPP pathway[27]. Pharmacological inhibition of PFKFB3 using AZ67 hampered the aberrant increase in neuronal glycolysis and alleviated the hallmarks of CLN7 disease pathogenesis after chronic in vivo intracerebroventricular administration. In contrast to neurons, astrocytes abundantly express PFKFB3[32], which in part explains the normal high glycolytic phenotype of these cells[27,32].

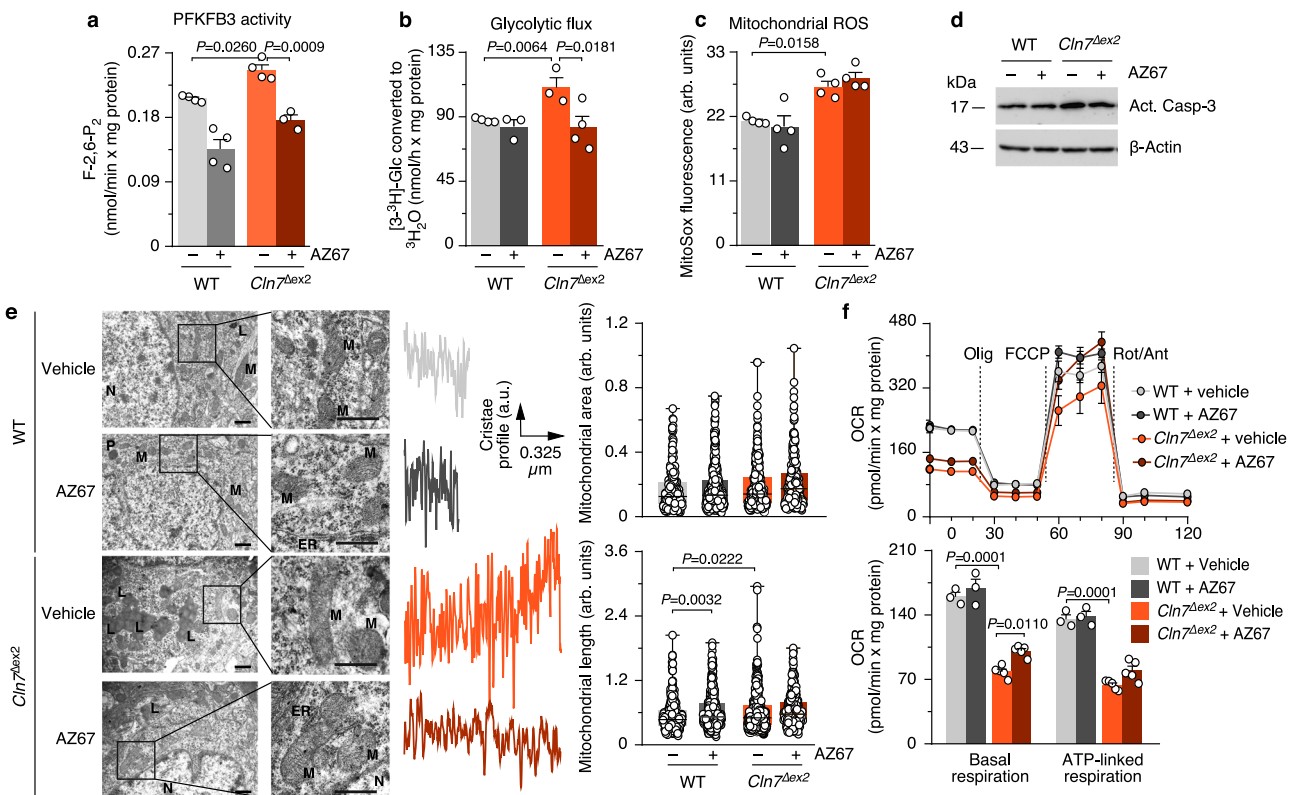

**Fig. 4 Pharmacological targeting PFKFB3 restores mitochondrial alterations of Cln7$^{\Delta ex2}$ disease in vivo. a–d** Analysis of PFKFB3 activity (**a**), glycolytic flux (**b**), mitochondrial ROS (**c**), and a representative active caspase-3 by western blot (**d**), in primary neurons incubated with the PFKFB3 inhibitor AZ67 (10 nM, 24 h). (ß-actin, loading control). Data are mean ± SEM from n = 4 independent experiments. **e** Representative electron microscopy images of the mouse brain cortex, after 2 months of a daily intracerebroventricular administration of PFKFB3 inhibitor AZ67 (1 nmol/mouse), displaying the cristae profile plot of intensities over the maximal axis of the magnified shown mitochondrion (left) and the analyses of mitochondrial area and length (right). Data are in box plots (the box extends from the 25th to 75th percentiles, the horizontal line indicates the median, and the whiskers go down to the smallest value and up to the largest) from n ≥ 179 mitochondria per condition of 5-month-old mice. Scale bars, 600 nm. (M mitochondria, L lysosome, ER endoplasmic reticulum, N nucleus). **f** OCR analysis (up) and calculated parameters (down) in primary neurons incubated with the PFKFB3 inhibitor AZ67 (10 nM, 24 h). Data are mean ± SEM from n = 3 (WT), n = 5 (Cln7$^{\Delta ex2}$) independent experiments. Statistical analyses performed by one-way ANOVA followed by Tukey's (**a**, **c**, **e**, **f**) or DMS's (**b**) post hoc tests. See also Supplementary Figs. 4 and 5. Source data are provided as a Source Data file.

At the dose of AZ67 administered, we show that glycolysis is inhibited in neurons, but unaltered in astrocytes[42] hence indicating that the main in vivo PFKFB3 target is neuronal. According to the partial recovery of respiration, and to the reduction in the cristae profile amplitude of mitochondria in the PFKFB3-inhibited Cln7$^{\Delta ex2}$ neurons, the protection exerted by PFKFB3 inhibition represents an adaptation of mitochondrial shape to a more bioenergetically efficient configurations[24,25]. Abnormal accumulation of mitochondria has also been reported in several forms of lysosomal-storage diseases[46], although their functional characterization is missing and the impact on other Batten disease pathogenesis unknown. Notably, mitochondrial membranes are required for autophagosomal biogenesis[47], an observation that opens the possibility that dysfunctional mitochondria may be a contributing factor in autophagy failure in CLN7 disease. In this context, it would be interesting to ascertain whether the bioenergetic alterations herein described in CLN7 disease are shared with other NCLs. If so, pharmacological inhibition of PFKFB3 would be a suitable therapeutic approach worth testing to delay and/or palliate the devastating consequences of each type of currently intractable[48] Batten disease.

## Methods

**Animals**. All protocols were performed according to the European Union Directive 86/609/EEC and Recommendation 2007/526/EC, regarding the protection of animals used for experimental and other scientific purposes, enforced in Spanish

legislation under the law 6/2013. Protocols were approved by the Bioethics Committee of the University of Salamanca or CIC bioGUNE ([$^{18}$F]FDG-PET and [$^{1}$H] MRS). Animals were bred at the Animal Experimentation Facility of the University of Salamanca in cages (maximum of five animals per cage), and a light–dark cycle was maintained for 12 h. The humidity was 45–65%, and the temperature was 20–25 ºC. Animals were fed ad libitum with a standard solid diet (17% proteins, 3% lipids, 58.7% carbohydrates, 4.3% cellulose, 5% minerals, and 12% humidity) and given free access to water. Cln7 knockout mouse carrying the European Conditional Mouse Mutagenesis (EUCOMM) tm1d allele by Cre-mediated recombination of the floxed exon 2 of the murine Cln7/Mfsd8 gene (Cln7$^{\Delta ex2}$)[12] were used. To abrogate mitochondrial ROS selectively in neurons in the Cln7$^{\Delta ex2}$ mice in vivo, we crossed Cln7$^{\Delta ex2}$ mice with transgenic mice harboring the full-length cDNA encoding catalase fused to the cytochrome c oxidase subunit VIII–mitochondrial leading sequence (mitoCatalase or mCAT), which has incorporated a floxed transcriptional STOP cassette between the mitochondrial-tagged catalase cDNA and the CAG promoter, which were previously generated in our laboratory by homologous recombination in the Rosa26 locus under a C57BL/6 background (mCAT$^{LoxP}$/+) in order to achieve tissue- and time-specific expression of mCAT in vivo[23]. mCAT$^{LoxP}$/+ mice were mated with mice harboring Cre recombinase under control of the neuronal-specific CAMKII promoter (CAMKIIa$^{Cre}$). The progeny, namely CAMKIIa$^{Cre}$/+; mCAT$^{LoxP}$/+, were crossed with Cln7$^{\Delta ex2}$/ Cln7$^{\Delta ex2}$ mice[12]. The offspring were crossed to obtain the following littermates genotypes: i) +/+; mCAT/+; +/+ (mCAT$^{LoxP}$); ii) +/+; mCAT/+; CamKIIa/+ (CAMKIIa$^{Cre}$-mCAT); iii) Cln7$^{\Delta ex2}$/Cln7$^{\Delta ex2}$; mCAT/+; +/+ (Cln7$^{\Delta ex2}$- mCA-T$^{LoxP}$); iv) Cln7$^{\Delta ex2}$/Cln7$^{\Delta ex2}$; mCAT/+; CamKIIa/+ (Cln7$^{\Delta ex2}$-CAMKIIa$^{Cre}$-mCAT).

**Genotyping by polymerase chain reaction (PCR)**. For Cln7$^{\Delta ex2}$ genotyping, a PCR with the following primers was performed 5′-TGGTGCATTAATACAGT CCTAGAATCCAGG-3′, 5′-CTAGGGAGGTTCAGATAGTAGAACCC-3′, 5′-TT

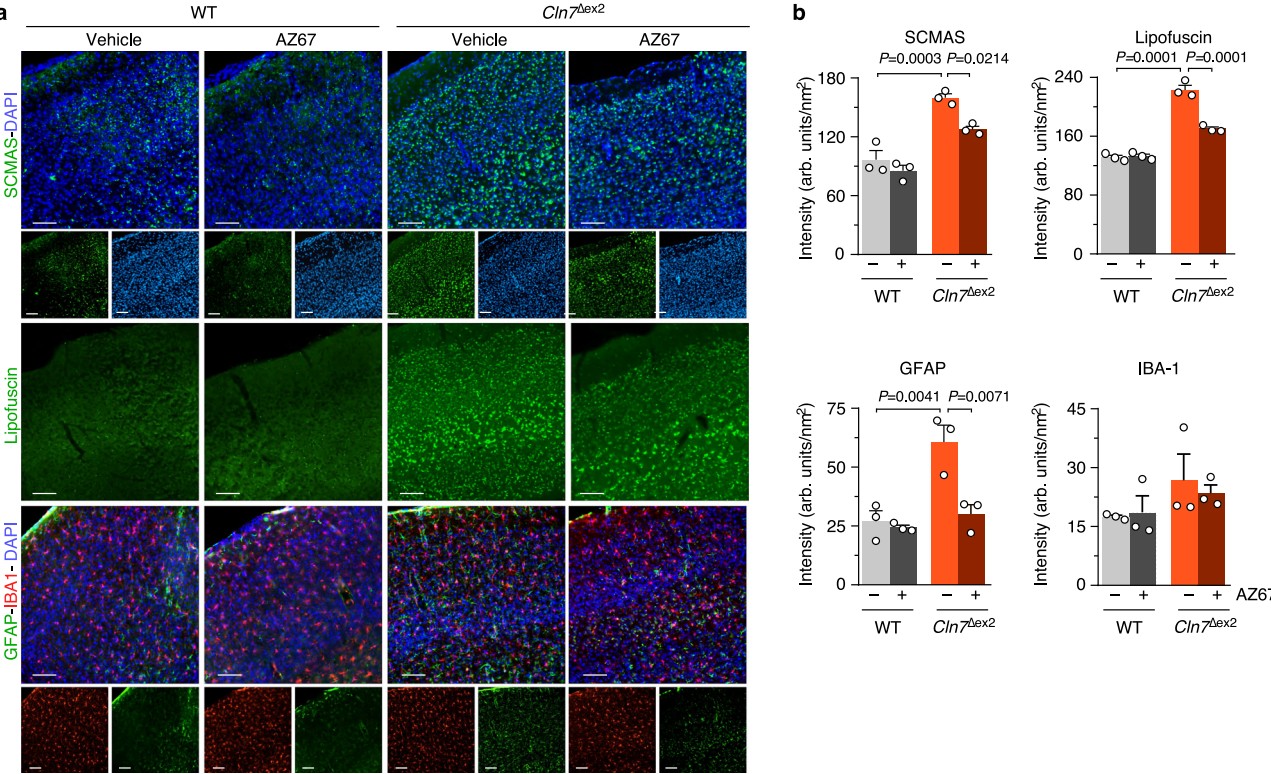

**Fig. 5 Pharmacological targeting PFKFB3 restores hallmarks of Cln7$^{\Delta ex2}$ disease in vivo. a** Representative images of SCMAS, lipofuscin, GFAP, and IBA-1 immunohistochemical analysis of the mouse brain cortex after 2 months of a daily intracerebroventricular administration of PFKFB3 inhibitor AZ67 (1 nmol/mouse). **b** Quantification of the images shown in panel a. Data are mean ± SEM from n = 3 animals of 5-month old (three serial slices per mouse). Scale bar, 100 μm. Statistical analyses performed by one-way ANOVA followed by Tukey's post hoc test. See also Supplementary Figs. 4 and 5. Source data are provided as a Source Data file.

CCACCTAGAGAATGGAGCGAGATAG-3′, resulting in a 290 bp band in the case of Cln7$^{\Delta ex2}$ mice, and 400 bp for wild type[12]. The primer sequences for genotyping the mCAT$^{LoxP}$ allele were 5′-CTCCCAAAGTCGCTCTGAGTTGT-TATCA-3′, 5′-CGATTTGTGGTGTATGTAACTAATCTGTCTGG-3′ and 5′-GC AGTGAGAAGAGTACCACCATGAGTCC-3′, which yielded a 778-bp band for the wild-type allele and a 245-bp band for the mCAT$^{LoxP}$ allele[23]. CaMKIIa$^{Cre}$ transgene was detected by amplifying a 270 bp region of Cre recombinase by PCR. Forward and reverse oligonucleotides used were, respectively, 5′-GCATTTCTG GGGATTGCTTTA-3′ and 5′-CCCGGCAAAACAGGTAGTTA-3′. An internal control was used to detect false negatives using the endogenous the alpha-synuclein (SCNA) gene. Its forward and reverse oligonucleotides were, respectively, 5′-AT CTGGTCCTTCTTGACAAAGC-3′ and 5′-AGAAGACCAAAGAGCAAGTGA CA-3′, which generated a 150 bp band.

**Reverse transcription–real-time quantitative PCR (RT–qPCR).** This was performed in total RNA samples, purified from the primary culture of neurons using the GenElute Mammalian Total RNA Miniprep Kit (Sigma), following the manufacturer's protocol. Amplifications were performed in 100 ng of RNA, using Power SYBR Green RNA-to-CT 1-Step kit (Applied Biosystems). The primers were 5′-TTCTCAGGTTTTTGCGGAGAAC-3′ and 5′-GTGCACATGTATGAGCTGG CA-3′ for PFKFB3; 5′-CTGAGTCCGAATCAGGTGCAG-3′ and 5′-GTCCATGG GAAGATGTTCTGG-3′ for spliced X-box binding protein 1 (XBP1) (sXBP1); 5′-T GGCCGGGTCTGCTGAGTCCG-3′ and 5′-GTCCATGGGAAGATGTTCTGG-3′ for total XBP1; 5′-GGGTTCTGTCTTCCACTCCA-3′ and 5′-AAGCAGCAGAGT CAGGCTTTC-3′ for activating transcription factor 4 (ATF4); 5′-CCACCACACC TGAAAGCAGAA-3′ and 5′-AGGTGAAAGGCAGGGACTCA-3′ for C/EBP homologous protein (CHOP); 5′-TTCAGCCAATTATCAGCAAACTCT-3′ and 5′-TTTTCTGATGTATCCTCTTCACCAGT-3′ for binding immunoglobulin protein (BiP); 5′-CTACCTGCGAAGAGGCCG-3′ and GTTCATGAGCTGCCCAC TGA-3′ for endoplasmic reticulum degradation-enhancing alpha-mannosidase-like protein 1 (EDEM1) and 5′-CGATGCCCTGAGGCTCTTTT-3′ and 5′-CAACGTC ACACTTCATGATG-3′ for β-actin. The mRNA abundance of each transcript was normalized to the β-actin mRNA abundance obtained in the same sample. WT neurons were used as a control.

**In vitro Cre recombinase activity induction.** Infection with adenovirus carriers of Cre recombinase and empty adenovirus (Control) was used to induce mCAT

expression in primary culture of neurons conditionals for mCAT expression (mCat$^{LoxP}$ and Cln7$^{\Delta ex2}$- mCAT$^{LoxP}$). The virus, transduced at 10 MOI, was purchased to Gene Transfer Vector Core (University of Iowa). Transduction was performed 3 days before cell recollection, and viral particles were left in the cultures for 24 h.

**Primary cultures.** Primary cultures of cortical neurons were prepared from the offspring of 14.5 days pregnant mice from Cln7$^{\Delta ex2}$ [12], mCat$^{LoxP}$, Cln7$^{\Delta ex2}$-mCAT$^{LoxP}$ or +/+ (WT) genotypes[49]. Cells were seeded at 2.0 × 10$^5$ cells per cm$^2$ in different-sized plastic plates coated with poly-D-lysine (10 μg/mL) and incubated in Neurobasal-A (Life Technologies) supplemented with 5.5 mM of glucose, 0.25 mM of sodium pyruvate, 2 mM glutamine, and 2% (vol/vol) B-27 supplement (Life Technologies). At 72 h after plating, medium was replaced, and cells were used at day 7. Cells were incubated at 37 °C in a humidified 5% (vol/vol) CO$_2$-containing atmosphere. Immunocytochemistry against a neuronal (β-tubulin III: 1/300; T2200; Sigma), astrocytic (GFAP: 1/800; AB5541; Millipore), oligodendrocytes (O4; 1/300; from mouse hybridoma kindly donated by Isabel Fariñas' laboratory), and microglial marker (CD45; 1/200; 553076; BD) was performed in order to determine the purity of the neuronal cultures that is 99.02% neurons, 0.43% astrocytes, 0.11% oligodendrocytes, 0.13% microglia, and 0.31% other cells.

**Induced pluripotent stem cells (iPSC) and neural progenitor cells (NPC) generation.** iPSC were generated from the dermal fibroblast of two CLN7 patients (Pa380 and Pa474) (approved by the UCL Research Ethics Committee), then characterized and differentiated to NPC as previously described[50]. Written informed consent was obtained from the patients. Human iPSC-derived NPCs from a control patient and patients Pa380 (c.881 C > A; pT294K; female of 2.5 years old with mental and speech regression, motor impairment without myoclonus or visual failure) and Pa474 (c.1393 C > T; p.R465W; male of 4.5-year old with motor impairment, mental and speech regression and myoclonus) harboring the indicated CLN7 homozygous mutations, were plated on Matrigel® Matrix in Nunc™ Lab-Tek™ eight-well Chamber Slides and cultured in neural expansion medium (NEM) with DMEM/F12, NEAA, N-2 supplement, B-27 supplement, heparin, bFGF protein, penicillin/streptomycin. iPSCs pluripotency was confirmed by immunocytochemistry using OCT4 (1/200) (ab19857; Abcam), SOX2 (1/100) (AF2018; R&D Systems), Nanog (1/100) (ab21624; Abcam) and Tra-1-60 (1/200) (MAB1295; R&D Systems), and by confirming their ability to differentiate into

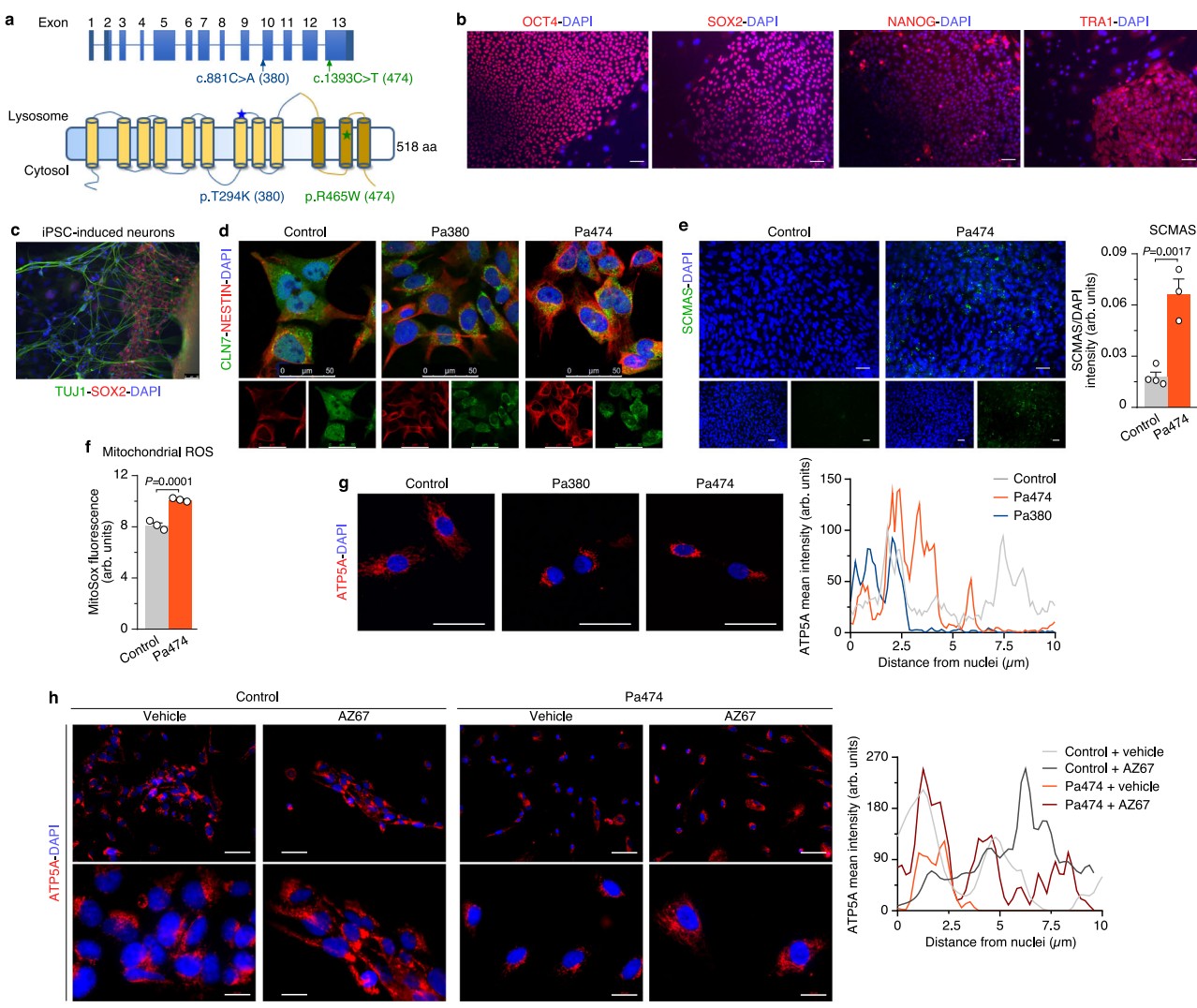

**Fig. 6 PFKFB3 inhibition in CLN7 patient-derived neural precursor cells restores mitochondrial condensation. a** Schematic representation of the locations of the CLN7 mutations found in patient 380 (Pa380, c.881 C > A; pT294K) and patient 474 (Pa474, c.1393 C > T; p.R465W). **b** iPSC characterization in Pa474 with the pluripotency markers OCT4, SOX2, Nanog, and Tra-1-60. Scale bar, 50 μm. **c** Characterization of differentiated neurons derived from iPSC in Pa474. Scale bar, 50 μm. **d** NPCs characterization in Pa380, Pa474, and a healthy, age-matched control individual. Scale bar, 50 μm. **e** Immunocytochemical analysis of SCMAS abundance in NPCs derived from Pa474 iPSC. Data are the mean ± SEM values from $n = 4$ (control), $n = 3$ (Pa474) independent samples (two-tailed Student's $t$ test). Scale bar, 50 μm. **f** Mitochondrial ROS analysis in NPCs. Data are the mean ± SEM values from $n = 3$ independent samples (two-tailed Student's $t$ test). **g** Immunocytochemical analysis of mitochondrial marker ATP5A in NPCs derived from Pa380, Pa474, and healthy-matched control patients. Scale bar, 50 μm. The right panel shows a representative pixel intensity profile of ATP5A across the maximal axis of the cell that departs from the nucleus. **h** Representative image of NPCs derived from Pa474 iPSC incubated with AZ67 for 24 h, fixed and subjected to immunocytochemical analysis for ATP5A. Scale bars, 60 μm (upper images of each condition) and 20 μm (lower images of each condition). The right panel shows a representative pixel intensity profile of ATP5A across the maximal axis of the cell that departs from the nucleus. Source data are provided as a Source Data file.

neurons using TUJ-1 (1/200) (MAB1195; R&D Systems) staining. NPC identity was confirmed by Nestin⁺/SOX2⁻ immunostaining (Nestin 1/100; MA1-110; Thermo Fisher).

**Freshly purification of neurons from the brain from adult mice.** Adult mouse brain (from 6-month-old animals) tissue was dissociated with the Adult Brain Isolation Kit (Miltenyi). Dissociated cells, after removal of debris and red blood cells, neurons were separated with the Neuron Isolation Kit (Miltenyi). The identity of the isolated fraction was confirmed previously[19] by western blot against the neuronal marker microtubule-associated protein 2 (MAP2) and GFAP.

**Cell treatments.** Neurons in primary culture were incubated with the rationally designed, potent, and highly selective PFKFB3 inhibitor AZ PFKFB3 67 (herein referred as AZ67)[41] (Tocris) (10 nM) or the calpain inhibitor MDL-28170 (MDL, 100 μM; Sigma) for 24 h. The cell-permeable $Ca^{2+}$-quelator BAPTA was used in the primary culture of neurons in the presence of Hanks's solution without calcium

(134.2 mM NaCl, 5.26 mM KCl, 0.43 mM $KH_2PO_4$, 4.09 mM $NaHCO_3$, 0.33 mM $Na_2HPO_4 \cdot 2H_2O$, 5.44 mM glucose, 20 mM HEPES, pH 7.4) for 1 h (10 μM; Sigma). NPCs were incubated with AZ67 (10 nM) for 24 h in NEM.

**Autophagy measurement.** To analyze the autophagy pathway, primary neurons were incubated in the absence or presence of the inhibitors of the lysosomal proteolysis leupeptin (100 μM) and ammonium chloride (20 mM) for 1 h. Cells were lysed and immunoblotted against LC3-II to assess autophagy, and against SCMAS and HSP60 to assess mitophagic flux[15].

**Cell transfections.** For knockdown experiments, small interfering RNA (siRNA) against CDK5 (siCDK5) (s201147; Thermo Fisher) was used. An siRNA control (siControl) (4390843; Thermo Fisher) was used in parallel. siRNA transfections were performed using the Lipofectamine RNAiMAX reagent (Thermo Fisher) at an siRNA final concentration of 9 nM. A solution of lipofectamine in OptiMEM medium (1:16, vol/vol) was mixed with the siRNAs, previously diluted in

OptiMEM (0.2 pmol/μl). This mixture was incubated 5 min at room temperature and then added to cells. Cells were used after 3 days.

**Total membrane purification**. Membranes were isolated from primary cultures of neurons, or whole-brain homogenates[12]. Cells or tissue were homogenized in sucrose buffer (200 mM sucrose, 50 mM Tris-HCl; pH 7.5, 1 mM EDTA) and centrifuged 5 minutes at $1500 \times g$. Supernatants were centrifuged at $20,000 \times g$ 10 min. Pellet was then homogenized in extraction buffer (50 mM Tris-HCl; pH 7.5, 1% (vol/vol) Triton-X-100, 1 mM EDTA) and incubated 30 min on ice, followed by a centrifugation at $20,000 \times g$ 10 min. Membranes enriched fraction remained in the supernatant.

**Western blotting**. Cells were lysed in RIPA buffer (1% sodium dodecylsulfate, 10 mM ethylenediaminetetraacetic acid (EDTA), 1% (vol/vol) Triton X-100, 150 mM NaCl, and 10 mM $Na_2HPO_4$, pH 7.0), supplemented with protease inhibitor mixture (Sigma), 100 μM phenylmethylsulfonyl fluoride, and phosphatase inhibitors (1 mM o-vanadate). Samples were boiled for 5 min. Aliquots of cell lysates (40 μg of protein) were subjected to SDS/PAGE on an 8–12% (vol/vol) acrylamide gel (MiniProtean; Bio-Rad) including PageRuler Prestained Protein Ladder (Thermo). The resolved proteins were transferred electrophoretically to nitrocellulose membranes (0.2 μm, Bio-Rad). Membranes were blocked with 5% (wt/vol) low-fat milk in TTBS (20 mM Tris, 150 mM NaCl, and 0.1% (vol/vol) Tween 20, pH 7.5) for 1 h. Subsequent to blocking, membranes were immunoblotted with primary antibodies overnight at 4 °C. After incubation with horseradish peroxidase-conjugated goat anti-rabbit IgG (1/10,000, Santa Cruz Biotechnologies), goat anti-mouse IgG (1/10,000, Bio-Rad), rabbit anti-goat IgG (1/10,000, Abcam) or goat anti-rabbit IgG (1/3000, Bio-Rad), membranes were immediately detected with the enhanced chemiluminescence kit WesternBright ECL (Advansta), or SuperSignal West Femto (Thermo) before exposure to Fuji Medical X-Ray film (Fujifilm), and the autoradiograms were scanned. Ponceau staining (Sigma) was occasionally used as an indicator of loading. At least three biologically independent replicates were always performed, although only one representative western blot is shown in the main figures. The protein abundances of all western blots per condition were measured by densitometry of the bands on the films using ImageJ 1.48u4 software (National Institutes of Health) and were normalized per the loading control protein. The resulting values were used for the statistical analysis. An uncropped scan of western blots present in the figures, with the replicas, can be found in Source Data file.

**Primary antibodies for western blotting**. Immunoblotting was performed with anti- C-subunit of ATP synthase (SCMAs) (1/1000) (ab181243; Abcam), anti-VDAC (1/666) (PC548; Calbiochem), anti-heat-shock protein-60 (HSP60) (1/666) (ab46798; Abcam), anti-PINK1 (1/500) (sc-33796; Santa Cruz Biotechnology), anti-NDUFS1 (1/500) (sc-50132; Santa Cruz Biotechnology), anti-CDK5 (1/500) (sc-6247; Santa Cruz Biotechnology), anti-PFKFB3 (1/500) (H00005209-M08; Novus Biologicals), anti-p25/35 (1/500) (2680; Cell Signalling), anti-caspase-3 (1/2000) (9661S; Cell Signalling), anti-CLN7 (1/500) (donated by Dr. Stephan Storch), anti-Parkin (1/100) (sc-32282; Santa Cruz Biotechnology), anti-LC3B (1/1000) (2775; Cell Signaling), anti-GFAP (1/500) (G6171; Sigma), anti-β-Tubulin III (1/500) (ab18207; Abcam) and anti-β-Actin (1/30,000) (A5441; Sigma).

**Immunocytochemistry**. Cells were seeded on coverslips, fixed with a 4% paraformaldehyde (PFA) solution, blocked, and incubated with primary antibodies overnight at 4 °C. The primary antibodies were anti-SCMAS (1/200) (ab181243; Abcam), anti-HSP60 (1/500) (ab46798; Abcam) and anti-LAMP1 (1/100) (1D4B; Developmental Studies Hybridoma Bank). They were then incubated for 1 h with fluorescent secondary antibodies (1/500) Alexa Fluor 488 anti-rabbit (A11008; Thermo Fisher), and Alexa Fluor anti-rat 647 (A-21247; Thermo Fisher). DAPI (4′,6-diamidino-2-phenylindole) was used for nuclei visualization. Coverslips were mounted in ProLong Gold antifade reagent. Negative controls were performed with either no primary or no secondary antibodies. No staining was detected in any case. Images were acquired on an Operetta CLS high-content imaging system (PerkinElmer) using ×63 (1.15 numerical aperture) objective. Images were acquired at the same exposure times in the same imaging session. Image quantification was performed after appropriate thresholding using the ImageJ software (NIH). The percentage of colocalization was calculated using the JAva COnstraint Programming (JACoP) plugin, specifically Manders' Overlap Coefficient, in single Z-stack sections.

**Mitochondrial isolation**. To obtain the mitochondrial fraction, cell pellets or brain cortex were frozen at −80 °C and homogenized (10–12 strokes) in a glass Teflon Potter–Elvehjem homogenizer in buffer A (83 mM sucrose and 10 mM MOPS; pH 7.2). The same volume of buffer B (250 mM sucrose and 30 mM MOPS) was added to the sample, and the homogenate was centrifuged ($1000 \times g$, 5 min) to remove unbroken cells and nuclei. Centrifugation of the supernatant was then performed ($12,000 \times g$, 3 min) to obtain the mitochondrial fraction, which was washed in buffer C (320 mM sucrose; 1 mM EDTA, and 10 mM Tris-HCl; pH 7.4)[19]. Mitochondria were suspended in buffer D (1 M 6-aminohexanoic acid and 50 mM Bis-Tris-HCl, pH 7.0).

**Blue-native gel electrophoresis and in-gel activity for complex I**. For the assessment of complex I organization, digitonin-solubilized (4 g/g) mitochondria (10–50 μg) were loaded in NativePAGE Novex 3–12% (vol/vol) gels (Life Technologies). After electrophoresis, in-gel NADH dehydrogenase activity was evaluated allowing the identification of individual complex I and complex I-containing supercomplexes bands due to the formation of purple precipitated at the location of complex I[19]. Briefly, gels were incubated in 0.1 M of Tris-HCl buffer (pH 7.4), 1 mg/ml of nitro blue tetrazolium, and 0.14 mM of NADH. Next, a direct electrotransfer was performed followed by immunoblotting against mitochondrial complex I antibody NDUFS1. The direct transfer of BNGE was performed after soaking the gels for 20 min (4 °C) in carbonate buffer (10 mM $NaHCO_3$; 3 mM $Na_2CO_3 \cdot 10H_2O$; pH 9.5–10). Proteins transfer to polyvinylidene fluoride (PVDF) membranes was carried out at 300 mA, 60 V, 1 h at 4 °C in carbonate buffer.

**Determination of PPP and glycolytic fluxes**. These were measured in 8-cm² flasks of primary cultures of neurons containing a central microcentrifuge tube with either 0.8 ml benzethonium hydroxide (Sigma) for $^{14}CO_2$ equilibration or 1 ml $H_2O$ for $^{3}H_2O$ equilibration. Incubations were carried out in KRPG (NaCl 145 mM; $Na_2HPO_4$ 5.7 mM; KCl 4.86 mM; $CaCl_2$ 0.54 mM; $MgSO_4$ 1.22 mM; pH 7.35) containing 5 mM D-glucose at 37 °C in the air-thermostatized chamber of an orbital shaker. To ensure adequate oxygen supply for oxidative metabolism throughout the incubation period, flasks were filled with oxygen (5% $CO_2/O_2$) before being sealed. To measure the carbon flux from glucose through the PPP, cells were incubated in KRPG (5 mM D-glucose) buffer supplemented with 0.5 μCi D-[1-$^{14}$C]glucose or [6-$^{14}$C]glucose for 90 min[27,51]. Incubations were terminated by the addition of 0.2 ml 20% perchloric acid (Merck Millipore), and 40 min before the benzethonium hydroxide (containing $^{14}CO_2$) was removed, and the radioactivity was measured with a liquid scintillation analyzer (Tri-Carb 4810 TR, PerkinElmer). PPP flux was calculated as the difference between $^{14}CO_2$ production from [1-$^{14}$C]glucose (which decarboxylates through the 6-phosphogluconate dehydrogenase-catalyzed reaction) and that of [6-$^{14}$C]glucose (which decarboxylates through the TCA cycle)[27,52]. Glycolytic flux was measured by assaying the rate of $^{3}H_2O$ production from [3-$^{3}$H]glucose through a similar method, but incubating cells with 3 μCi D-[3-$^{3}$H]glucose in KRPG buffer per flask for 120 min[27,51]. Incubations were terminated with 0.2 ml 20% perchloric acid, and the cells were further incubated for 96 h to allow for $^{3}H_2O$ equilibration with $H_2O$ present in the central microcentrifuge tube. The $^{3}H_2O$ was then measured by liquid scintillation counting (Tri-Carb 4810 TR, PerkinElmer). Under these experimental conditions, 75% of the produced $^{14}CO_2$ or 28% of the produced $^{3}H_2O$ was recovered and used for the calculations[51].

**Lactate determination**. Lactate concentrations were measured in the culture medium espectrophotometrically[27] by determination of the increments in the absorbance of the samples at 340 nm in a mixture containing 1 mM $NAD^+$, 8.25 U lactate dehydrogenase in 0.25 M glycine, 0.5 M hydrazine, and 1 mM EDTA buffer, pH 9.5.

**Fructose-2,6-bisphosphate determinations**. For F-2,6-$P_2$ determinations, cells were lysed in 0.1 M NaOH and centrifuged ($20,000 \times g$, 20 min). An aliquot of the homogenate was used for protein determination, and the remaining sample was heated at 80 °C (5 min), centrifuged ($20,000 \times g$, 20 min) and the resulting supernatant was used for the determination of F-2,6-$P_2$ concentrations using a coupled enzymatic reaction[53]. This approach reveals the relative abundance of F-2,6-$P_2$ generated by PFKFB3 by the coupled enzymatic activities of PFK1 (Sigma) (in the presence of 1 mM fructose-6-phosphate and 0.5 mM pyrophosphate), aldolase (Sigma), and triose-phosphate isomerase/glycerol-3-phosphate dehydrogenase (Sigma). This reaction generates glycerol-3-phosphate and oxidizes NADH (Sigma), producing a reduction in the absorbance at 340 nm that is monitored spectrophotometrically.

**Phos-tag SDS-PAGE**. For the evaluation of phosphorylation levels of CDH1, primary cultures of neurons were homogenized in extraction buffer (100 mM NaCl, 50 mM Tris pH 8, 1% (vol/vol) NP40). Electrophoresis was performed in 8% (vol/vol) SDS-PAGE gels in the presence of 37.5 μM of PhosTag Acrylamide (ALL-107M, Wako) and 75 μM of $MnCl_2$. After electrophoresis, gels were washed three times in transfer buffer with 1 mM of EDTA, before electroblotting.

**Mitochondrial ROS**. Mitochondrial ROS were determined with the fluorescent probe MitoSox (Life Technologies). Neurons, from primary cultures or adult brain-cell suspensions, were incubated with 2 μM of MitoSox for 30 min at 37 °C in a 5% $CO_2$ atmosphere in HBSS buffer (134.2 mM NaCl, 5.26 mM KCl, 0.43 mM $KH_2PO_4$, 4.09 mM $NaHCO_3$, 0.33 mM $Na_2HPO_4 \cdot 2H_2O$, 5.44 mM glucose, 20 mM HEPES and 20 mM $CaCl_2 \cdot 2H_2O$, pH 7.4). The cells were then washed with phosphate-buffered saline (PBS: 136 mM NaCl; 2.7 mM KCl; 7.8 mM $Na_2HPO_4 \cdot 2H_2O$; 1.7 mM $KH_2PO_4$; pH 7.4) and collected by trypsinization. MitoSox fluorescence intensity was assessed by flow cytometry (FACScalibur flow cytometer, BD Biosciences) and expressed in arbitrary units. A schematic representation of the gating strategy can be found in Supplementary Fig. 6a.

**H$_2$O$_2$ determination**. For H$_2$O$_2$ assessments, AmplexRed (Life Technologies) was used. Cells were trypsinized and incubated in KRPG buffer (145 mM NaCl, 5.7 mM Na$_2$HPO$_4$, 4.86 mM KCl, 0.54 mM CaCl$_2$, 1.22 mM MgSO$_4$, 5.5 mM glucose, pH 7.35) in the presence of 9.45 μM AmplexRed containing 0.1 U/mL horseradish peroxidase. Luminescence was recorded for 2 h at 30 min intervals using a Varioskan Flash (Thermo Scientific) (excitation, 538 nm; emission, 604 nm). Slopes were used for calculations of the rates of H$_2$O$_2$ formation.

**Mitochondrial membrane potential**. The mitochondrial membrane potential ($\Delta\psi_m$) was assessed with MitoProbe DiIC$_1$(5) (Life Technologies) (50 nM) by flow cytometry (FACScalibur flow cytometer, BD Biosciences) and expressed in arbitrary units. For this purpose, cell suspensions were incubated with the probe 30 min at 37 °C in PBS. $\Delta\psi_m$ are obtained after subtraction of the potential value determined in the presence of carbonyl cyanide-4-(trifluoromethoxy)phenylhydrazone (CCCP) (10 μM, 15 min) for each sample. A schematic representation of the gating strategy can be found in Supplementary Fig. 6b.

**Cytosolic Ca$^{2+}$ determination using Fura-2 fluorescence**. To estimate the intracellular Ca$^{2+}$ levels in neurons we used the fluorescent probe Fura-2 (acetoxymethyl-derivative; Life Technologies)[54]. Neurons were incubated with Fura-2 (2 μM) for 40 min in neurobasal medium at 37 °C. Then, cells were washed and further incubated with standard buffer (140 mM NaCl, 2.5 mM KCl, 15 mM Tris-HCl, 5 mM D-glucose, 1.2 mM Na$_2$HPO$_4$, 1 mM MgSO$_4$ and 1 mM CaCl$_2$, pH 7.4) for 30 min and 37 °C. Finally, the standard buffer was removed and experimental buffer (140 mM NaCl, 2.5 mM KCl, 15 mM Tris-HCl, D-glucose, 1.2 mM Na$_2$HPO$_4$, and 2 mM CaCl$_2$, pH 7.4) was added. Emissions at 510 nm, after excitations at 335 and 363 nm, respectively, were recorded in a Varioskan Flash (Thermo) spectrofluorometer at 37 °C. Ca$^{2+}$ levels were estimated by representing the ratio of fluorescence emitted at 510 nm obtained after excitation at 335 nm divided by that at 363 nm (F335/F363). Background subtraction was accomplished from emission values obtained in Fura-2-lacking neurons. At least, 6 wells were recorded per condition in each experiment and the averaged values are shown, normalized per mg of protein present in the sample.

**Bioenergetics**. Oxygen consumption rates of neurons were measured in real-time in an XFe24 Extracellular Flux Analyzer (Seahorse Bioscience; Seahorse Wave Desktop software 2.6.1.56). The instrument measures the extracellular flux changes of oxygen in the medium surrounding the cells seeded in XFe24-well plates. The assay was performed on day 7 after cell plating/culture. Regular cell medium was then removed, and cells were washed twice with DMEM running medium (XF assay modified supplemented with 5 mM glucose, 2 mM L-glutamine, 1 mM sodium pyruvate, 5 mM HEPES, pH 7.4) and incubated at 37 °C without CO$_2$ for 30 min to allow cells to pre-equilibrate with the assay medium. Oligomycin, FCCP or antimycin/rotenone diluted in DMEM running medium were loaded into port-A, port-B, or port-C, respectively. Final concentrations in XFe24 cell culture microplates were 1 μM oligomycin, 2 μM FCCP, 2.5 μM antimycin, and 1.25 μM rotenone. The sequence of measurements was as follow unless otherwise described. The basal level of oxygen consumption rate (OCR) was measured three times, and then port-A was injected and mixed for 3 min after OCR was measured three times for 3 min. Same protocol with port-B and port-C. OCR was measured after each injection to determine the mitochondrial or non-mitochondrial contribution to OCR. All measurements were normalized to average three measurements of the basal (starting) level of cellular OCR of each well. Each sample was measured in three to five wells. Experiments were repeated three to five times with different cell preps. Non-mitochondrial OCR was determined by OCR after antimycin/rotenone injection. Maximal respiration was determined by maximum OCR rate after FCCP injection minus non-mitochondrial OCR. ATP production was determined by the last OCR measurement before oligomycin injection minus the minimum OCR measurement after oligomycin injection.

**Activity of mitochondrial complexes**. Cells were collected and suspended in PBS (pH 7.0). After three cycles of freeze/thawing, to ensure cellular disruption, complex I, complex II, complex II–III, complex IV, and citrate synthase activities were determined. Rotenone-sensitive NADH-ubiquinone oxidoreductase activity (complex I)[55] was measured in KH$_2$PO$_4$ (20 mM; pH 7.2) in the presence of 8 mM MgCl$_2$, 2.5 mg/mL BSA, 0.15 mM NADH, and 1 mM KCN. Changes in absorbance at 340 nm (30 °C) ($\varepsilon = 6.81$ mM$^{-1}$ cm$^{-1}$) were recorded after the addition of 50 μM ubiquinone and 10 μM rotenone. Complex II–III (succinate–cytochrome c oxidoreductase) activity[56] was determined in the presence of 100 mM phosphate buffer, plus 0.6 mM EDTA(K$^+$), 2 mM KCN, and 200 μM cytochrome c. Changes in absorbance were recorded (550 nm; 30 °C) ($\varepsilon = 19.2$ mM$^{-1}$ cm$^{-1}$) after the addition of 20 mM succinate and 10 μM antimycin A. For complex IV (cytochrome c oxidase) activity, the first-rate constant of cytochrome c oxidation was determined[57] in the presence of 10 mM phosphate buffer and 50 μM reduced cytochrome c; absorbance was recorded every minute at 550 nm, 30 °C ($\varepsilon = 19.2$ mM$^{-1}$ cm$^{-1}$). Citrate synthase activity[58] was measured in the presence of 93 mM Tris-HCl, 0.1% (vol/vol) Triton X-100, 0.2 mM acetyl-CoA, 0.2 mM DTNB; the reaction was started with 0.2 mM oxaloacetate, and the absorbance was recorded at 412 nm (30 °C) ($\varepsilon = 13.6$ mM$^{-1}$ cm$^{-1}$).

**Protein determinations**. Protein samples were quantified by the BCA protein assay kit (Thermo) using BSA as a standard.

**Stereotaxic cannula implantation**. For intracerebroventricular injections (icv) a cannula was placed[59,60]. Anesthetized mice with sevoflurane (Sevorane; Abbott) were placed in a stereotaxic frame (Model 1900; David Kopf Instruments) with a micromanipulator (Model 1940; David Kopf Instruments) and a digital reading system (Wizard 550; Anilam). The cranium was exposed to properly reach the coordinates for the lateral ventricle (coordinates from bregma: anteroposterior:−0,2; center-center: ±0,9; dorsoventral: −2)[61], where a small skull hole was performed with a drill (Model 1911; Kopf Instruments). The cannula was placed (C315G/SPC; Plastics-One) and fixed with glue (Loctite 454; Henkel) and dentist cement (Hiflex RR; PrevestDenPro). A dummy was placed (C315GS-5-SP; Plastics-One) to prevent dust brain contamination. Mice after surgery were kept above heating plates (Plactronic Digital; JP Selecta) and fed with soaked food until recovery (at least for 15 days).

**Pharmacokinetics of AZ67**. For the pharmacokinetic assay, healthy male C57BL/6 mice were used. A single dose of 40 mg/kg of AZ67 was injected intravenously and the blood, cerebrospinal fluid (CSF) and brain, were collected after 5 min, 15 min, 30 min, 1 h, 2 h, 4 h, 8 h, and 24 h. AZ67 concentrations in the different samples were determined by liquid chromatography followed by MS/MS[41].

**In vivo toxicity assay**. Male mice (C57BL6/J; six animals per group; 8-week old) (purchased from Charles-River, Spain) were subjected to the implantation of a cannula in the lateral ventricle under anesthesia and then left for at least 15 days for full recovery. After this, the PFKFB3 inhibitor (AZ67) was administered through the cannula using an automatic micro-pump (CMA 4004 Microdialysis Syringe Pump, Harvard Apparatus) at different doses: 0 (vehicle), 0.005, 0.01, 0.05, 0.1, 1, and 10 nmol/mice. The compounds were administered every 24 h for 1 week, and animals were analyzed in the open field immediately before each administration. We selected the maximal dose that caused no evident alterations and/or deterioration of the animals for the following experiments, being 1 nmol/mouse.

**Open-field tests**. Male mice were left to acclimate in the room for no less than 15 min at the same time of day (10:00 to 14:00). Tracking was carried out one at a time, and we carefully cleaned the apparatus with 70% ethanol between trials to remove any odor cues. An ANY-box core was used, which contained a light-gray base and an adjustable perpendicular stick holding a camera and an infrared photobeam array to track the animal movement and to detect rearing behavior, respectively. Mouse movements were tracked with the ANY-maze 5.33 software and the ANY-maze interface to register all parameters described subsequently. For the open-field test, a 40 × 40 × 35 cm (w, d, h), black infrared transparent Perspex insert was used, and the arena was divided into three zones, namely border (8 cm wide), center (16% of the total arena) and intermediate (the remaining area). The test lasted for 10 min, and the distance traveled, and the time spent in each zone was measured.

**AZ67 in vivo administration**. AZ67 (Tocris) for in vivo usage was dissolved in 20% (wt/vol) PEG200 in PBS to a 20 mM concentration. Four groups were generated (four to six animals/group), namely: WT-vehicle, Cln7$^{\Delta ex2}$-vehicle, WT-AZ67, Cln7$^{\Delta ex2}$-AZ67. The cannula was inserted intracerebroventricularly at the age of 8 weeks and, after at least 15 days of recovery, we injected the AZ67 at the dose identified previously (1 nmol/mouse) every 24 h. The duration of the experiment was determined by the presence of hindlimb clasping the Cln7$^{\Delta ex2}$ vehicle-treated mice, being this time two months. After this, the animals were perfused, and their brains dissected to be investigated by immunofluorescence and electron microscopy.

**Electron microscopy and mitochondrial morphology analysis**. Male mice were anaesthetized by intraperitoneal injection of a mixture of xylazine hydrochloride (Rompun; Bayer) and ketamine hydrochloride/chlorbutol (Imalgene; Merial) (1:4) at 1 ml per kg body weight and then perfused intra-aortically with 0.9% NaCl followed by 5 ml/g body weight of 2% (wt/vol) paraformaldehyde plus 2% (vol/vol) glutaraldehyde. After perfusion, brains were dissected out sagitally in two parts and post-fixed with perfusion solution overnight at 4 °C. Brain blocks were rinsed with 0.1 M PB solution and a 1 mm$^3$ squared of brain cortex was excised and treated with osmium tetroxide (1% in PB) for 1 h. Tissue was then washed with distilled water and dehydrated in ascending series of ethanol followed by embedment in EPON resin. Ultra-thin sections (50 nm) were stained with uranyl acetate and lead citrate and examined with Tecnai Spirit Twin 120 kv transmission electron microscopy equipped with a digital camera Orius WD or JEM-1010 (JEOL) 100 kv transmission electron microscopy equipped with a digital camera AMT RX80. For mitochondrial area quantification, the area of each mitochondrion was quantified in neuronal soma, axons and dendrites. In the case of mitochondrial length, the values represent the length in the maximal axis of mitochondria in the plane of microphotographies. Cristae profiles of representative mitochondria of each condition and type were traced along the major axis that crosses mitochondria

perpendicularly to cristae. Data of pixel intensity were obtained using the plot profile plugin of ImageJ software.

**Mouse perfusion and immunohistochemistry**. Mice (5 months for AZ67 intraventricular injections; 3 months for mCAT expression approach) were anaesthetized by intraperitoneal injection of a mixture of xylazine hydrochloride (Rompun; Bayer) and ketamine hydrochloride/chlorbutol (Imalgene; Merial) (1:4) at 1 ml per kg body weight and then perfused intra-aortically with 0.9% NaCl followed by 5 ml p/g body weight of Somogyi (4% (wt/vol) paraformaldehyde, and 0.2% (vol/vol) picric acid, in 0.1 M PB; pH 7.4). After perfusion, brains were dissected out sagitally in two parts and post-fixed with Somogyi for 2 h at room temperature. Brain blocks were rinsed successively for 10 min, 30 min and 2 h with 0.1 M PB solution and cryoprotected in 10%, 20 and 30% (wt/vol) sucrose in PB sequentially, until they sank. After cryoprotection, 40-μm-thick sagittal sections were obtained with a freezing–sliding cryostat (Leica). Sectioning of WT and $Cln7^{\Delta ex2}$ brains were performed under the same conditions and sessions. The sections were collected serially in a 12-well plate in 0.1 M PB, rinsed three times for 10 min in 0.1 M PB, and used for subsequent immunohistochemistry and lipofuscin observation. The section-containing wells that were not used were kept in a freezer mix (30% (vol/vol) polyethylene glycol, 30% (vol/vol) glycerol in 0.1 M PB) at −20 °C. For immunohistochemistry, sections were incubated sequentially in (i) 5 mg/ml sodium borohydride in PB for 30 min (to remove aldehyde autofluorescence); (ii) three PBS washes of 10 min each; (iii) 1/500 anti-GFAP (G6171; Sigma) and 1/500 anti-IBA-1 (019–19741; Wako) or 1/500 anti-ATP-C (SCMAS) (ab181243; Abcam) in 0.02% Triton X-100 (Sigma) and 5% goat serum (Jackson Immuno-Research) in 0.1 M PB for 72 h at 4 °C; (iv) three PB washes of 10 min each; (v) fluorophore-conjugated secondary antibodies, 1/500 Cy2 goat anti-mouse and 1/500 Cy3 goat anti-rabbit (Jackson Immuno-Research) or Alexa-488 (A11008; Molecular Probes) or 1/800 Cy3 donkey anti-rat (Jackson Immuno-Research) in PB for 2 h at room temperature; and (vi) 0.5 μg/ml DAPI in PB for 10 min at room temperature. After being rinsed with PB, sections were mounted with Fluoromount (Sigma) or Fluorosave (Millipore) aqueous mounting medium and coverslips (Thermo Fisher)[62]. For autofluorescence (lipofuscine accumulation), sections were mounted directly.

**Imaging and quantification**. Sections were examined with epifluorescence and the appropriate filter sets under an Operetta CLS high-content imaging system (PerkinElmer). Large fields of view were acquired with an ×5 scan using an OperaPHX/OPRTCLS ×5 Air Objective. Then high-resolution images were acquired using an OperaPHX/OPRTCLS Air Objective ×20 hNA objective. Immunohistochemical digital images were used to analyze different protein staining in the three most sagittal sections per animal. Images were analyzed with the Harmony software with PhenoLOGIC (PerkinElmer). Interest brain areas (cortex, hippocampus, and cerebellum) were selected and subsequently quantified as mean intensity per area by using the "measure rectangle" function, which represents the mean intensity of a channel per selected area.

**NPC immunocytochemistry**. NPCs were fixed with 100% ice-cold methanol for 5 min and incubated in blocking solution (1% (vol/vol) normal goat serum, 0.1% (wt/vol) bovine serum albumin (BSA), 0.1% (vol/vol) Triton X-100 in DPBS). The antibodies were incubated in a blocking solution. The incubation of the primary antibody (anti-ATP5A, (1/100) (ab14748; Abcam) or SCMAS (1/200) (ab181243; Abcam) was performed for 2 h at room temperature, and the secondary antibodies (Alexa Fluor 568 goat anti-mouse or Alexa Fluor 488 goat anti-rabbit (1/500)) were applied for 1 h at room temperature. Slides were mounted with VECTASHIELD Mounting Medium with DAPI, incubated for 24 h at 4 °C, and imaged with a Zeiss Axio Imager M2 fluorescence microscope or under an inverted microscope (Nikon; Eclipse Ti-E) equipped with a pre-centered fiber illuminator (Nikon; Intensilight C-HGFI), B/W CCD digital camera (Hamamatsu; ORCA-E.R.). Fluorescence quantification was performed after appropriate thresholding using the ImageJ software (NIH). The pixel intensity profile of ATP5A immunodecoration was analyzed across the maximal axis of the cell that departs from the nucleus, using the plot profile plugin of ImageJ software. A representative profile is shown for each condition.

**Positron-emission tomography (PET)**. 2-[18F]fluoro-2-deoxy-D-glucose ([18F]FDG) was kindly donated by Curium Pharma Spain (FLUORSCAN 3000). Before imaging studies, animals (5-month old) were fastened for 6 h with free access to drinking water. Administration of [18F]FDG (19.2 ± 1.6 MBq, 100 μL) was carried out via one of the lateral tail veins under anesthesia, induced with 3.0–5.0% isoflurane in pure oxygen and maintained with 1.5–2.0% isoflurane in pure oxygen. After administration, animals were allowed to recover from anesthesia for 45 minutes before being subjected to positron-emission tomography (PET) studies. PET studies ($n = 5$ for control and study groups; 10 min acquisitions) were conducted using the β-cube microsystem (Molecubes, Gent, Belgium), with the head of the animal positioned in the center of the field of view, in one-bed position using a 511 keV ± 30% energetic window. A computerized tomography (CT) scan was acquired immediately after the finalization of the PET imaging session, both for anatomical reference and to determine the attenuation map for PET image

reconstruction. PET images were reconstructed with OSEM-3D iterative algorithm. Images were analyzed using π-MOD image analysis software (π-MOD Technologies Ltd, Zurich, Switzerland). With that aim, PET images were manually co-registered with a M. Mirrione-T2 MRI atlas available at π-MOD software. Volumes of interest (VOIs) were automatically delineated in different brain regions, namely cortex, cerebellum, brain stem, hippocampus, striatum, and whole brain, and the concentration of radioactivity in each region was determined and decay-corrected to injection time. Values were finally normalized to injected amount of radioactivity and body weight, to obtain standard uptake values (SUVs).

**Magnetic resonance spectroscopy**. Localized [1]H-MRS was performed at 11.7 Tesla using a 117/16 USR Bruker Biospec system (Bruker Biospin GmbH, Ettlinglen, Germany) interfaced to an advance III console and operating ParaVicion 6.1 under topspin software (Bruker Biospin). After fine-tuning and shimming of the system, water-signal FWHM values typically in the 15–25 Hz range were achieved. Scanning started with the acquisition of three scout images (one coronal, one transverse, and one sagittal) using a 2D-multiplane T2W RARE pulse sequence with Bruker's default parameters. Those images were used to place the spectroscopy voxel of size $1.5 \times 1.5 \times 2$ mm³ located at the right striatum of the mouse brain or $2 \times 0.8 \times 2$ mm³ located in the cortex (at the mid-line of the brain), always with care not to include the ventricles in the voxel (the geometry of the voxel was slightly altered to avoid this event, when necessary). At least two [1]H-MRS spectra were acquired per scanning session per animal (5-month-old animals). The voxel was repositioned, and shimming adjustments were repeated between acquired spectra, when the spectral resolution of the obtained [1]H-spectrum was not good. For [1]H-MR a water suppressed PRESS sequence was used with the following parameters: Echo time = 17.336 ms (TE1 = TE2 = 8.668 ms); Repetition time = 2500 ms; Naverages = 256; Acquisition size = 2048 points; spectral width = 11 ppm (5498.53 Hz). MR spectra were fitted and quantified using LC-Model 6.3-1R[63].

**Statistical analysis**. The comparisons between two groups of values we performed using two-tailed Student's $t$ test. For multiple-values comparisons, we used one-way ANOVA followed by either Tukey or DMS post hoc tests, as indicated in the figure legends. The statistical analysis was performed using the GraphPad Prism v8 software. The number of biologically independent culture preparations or animals used per experiment are indicated in the figure legends, and the $P$ values in the figures.

**Reporting summary**. Further information on research design is available in the Nature Research Reporting Summary linked to this article.

## Data availability

The datasets generated and analyzed during the current study are available from the corresponding author on reasonable request. Source data are provided with this paper.

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

## Acknowledgements

We acknowledge the technical assistance of Monica Resch, Monica Carabias-Carrasco, Lucia Martin and Estefania Prieto-Garcia, from the University of Salamanca. This work was funded by the European Regional Development Fund, European Union's Horizon 2020 Research and Innovation Programme (BATCure grant No. 666918 to J.P.B., S.E.M., D.L.M., S.S., and T.R.M; PANA grant No. 686009 to A.A.), Agencia Estatal de Investigación (PID2019-105699RB-I00/AEI/10.13039/501100011033 and RED2018-102576-T to J.P.B; SAF2017-90794-REDT to A.A.), Instituto de Salud Carlos III (CB16/10/00282 to J.P.B.; PI18/00285; RD16/0019/0018 to A.A.), Junta de Castilla y León (CS/151P20 and Escalera de Excelencia CLU-2017-03 to J.P.B. and A.A.), Ayudas Equipos Investigación Biomedicina 2017 Fundación BBVA (to J.P.B.), and Fundación Ramón Areces (to J.P.B. and A.A.). SM benefits from MRC funding to the MRC Laboratory for Molecular Cell

Biology University Unit at UCL (award code MC_U12266B) towards lab and office space. Part of this work was funded by Gero Discovery L.L.C. M.G.M. is an ISCIII-Sara Borrel contract recipient (CD18/00203).

## Author contributions

Conceived the idea: J.P.B. Designed research: J.P.B., I.L.F., A.A., P.O.F., S.E.M., D.L.M., and T.M. Performed research: I.L.F., M.G.M., C.B., O.B., N.B., B.M.F., P.A.B., C.V.G., D.J.B., R.Q.C., E.F., J.L., P.R.C., A.S., M.G.F., L.F., and C.D.T. Analyzed the data: J.P.B., I.L.F., M.G.M., C.B., O.B., N.B., A.A., J.L., P.R.C., A.S., M.G.F., L.F., and T.M. Contributed materials: P.O.F. and S.S. Wrote the manuscript: J.P.B., I.L.F., and M.G.M. Edited and approved the manuscript: all co-authors.

## Competing interests

P.O.F. is a shareholder and O.B. is an employee of Gero Discovery LLC, a company developing PFKFB3 inhibitors. The remaining authors declare no competing interests.
