## [Peer Review File · Nature Communications]

Reviewers' Comments:

Reviewer #1:

Remarks to the Author:

The present manuscript examines a mouse model of neuronal ceroid lipofuscinosis. Although mitochondria are not my primary expertise, it seems to me that the evidence for mitochondrial dysfunction is good quality. The studies around glycolysis and the pentose phosphate pathway are intriguing but not fully persuasive: Changes in glycolytic flux are small, in some cases negligible (Even if nominally statistical fully significant), change in PFKFB3 levels are also small, making it hard to evaluate if the proposed main effects, much less mechanism, are valid. On the flipside, the pharmacological therapy is an impressive effort and the assays for measuring metabolism, while not very comprehensive, are generally well-chosen (Except for use of non-physiological conditions for the key radioactive assays) but not in vivo. I would suggest the following:

1. Given the main claim in the title, it seems that the Up regulation of glycolysis must be proven without a doubt. To this end, the authors need to address 4 deficiencies:

a) provide in vivo evidence for glycolysis upregulation. This is admittedly not easy, but would be very desirable. Consider PET or ^{13}C or ^2H glucose tracing.

b) Strengthen the in vitro evidence by (i) Providing some insight into why the effect sizes so much smaller in 3b than 3a. (ii) measure also glucose depletion from the medium over time. This can be done with a standard glucometer, but does require careful choice of medium glucose so that the decrease is observable.

c) Fix deficiency of radioactive experiments by repeating at least a subset of them in some version of a normal medium not PBS (which should not be required to make radioactive measurements)

d) For oxPPP measurements, the radioactive as is really the only reliable and also easy one, but authors need to repeat in normal medium and show separately $1\text{-}^{14}\text{C}$ and $6\text{-}^{14}\text{C}$ data in the supplement.

2) I would suggest changing "mediates" to "in" in the title, so the Authors have less pressure to prove that a major damage pathway goes through glycolysis, for which a single pharmacological manipulation is insufficient.

3) Put some of the key data in the supplement on lactate secretion and PFKFB3 quantitation into the maintext. The PFKFB3 blots are not persuasive but the quantitation shows a trend that is statistically significant across samples. Also, weaken claims by explicitly pointing out that these are small effect size in abstract and main text.

4) Change "pharmacological inhibition of PFKFB3" in abstract to the name of the agent as a single pharmacological agent is not sufficient to prove that the activity flows through the protein target without genetics, in vivo target engagement validation, or rescue, all of which are missing (or provide 1 of these).

5) Add "may" to 1st part of the conclusion sentence of abstract. "Aberrant upregulation of neuronal glycolysis MAY contribute to CLN7 pathogenesis"

6) Better explain methods for F26BP measurement, provide raw data supporting the measurement in the supplement, note that effect size here also is small

Reviewer #2:

Remarks to the Author:

In this study, Lopez-Fabuel et al. investigated that abnormal upregulation of glycolysis, which is due to PFKFB3 stabilization, contributes to CLN7 neuronal ceroid lipofuscinosis (NCLs).

I recommend the work can be published but the following queries need to be addressed.

1. In figure 3l, p35 western blot data is not accordant with supplementary figure 3i. The p35 p25 protein expression level in both Cln7 Δ ex2 groups (with or without MDL) looks like similar.

2. The authors need to explain why Cdk5 protein expression is upregulated in Cln7 Δ ex2 neurons. Have that phenomenon been well known in CLN7 neuronal ceroid lipofuscinosis? The increased Cdk5 protein expression may also affect hyperphosphorylation of Cdh1.

3. Please insert 'h' in the figure 3. Figure 3h is missing.

4. The authors should explain what Pa474 is in Figure 4h in the manuscript.
5. The manuscript text should be carefully edited. There are errors such as '...such as CLN2/TPP1 (Ref. 11), here ...' (p3, line 71), and '...a robust positive effector of PFK1 (Ref. 26). The rate...' (p7, line 158).
6. Please mention RRID of antibodies in the manuscript.

Reviewer #4:

Remarks to the Author:

General comments-

Using Cln7 Δ ex2 mice, a reliable animal model of CLN7-disease, Lopez-Fabuel and colleagues have conducted a comprehensive study to explore the mechanism of pathogenesis underlying CLN7 disease. Their results are impressive and clearly demonstrate that mitochondrial oxidative stress causing increased glycolysis, at least in part, contributes to CLN7 pathogenesis. They further demonstrate that the use of pharmacological agents targeting RKF3B3 may be a potential therapeutic target for this disease. While their investigations are thorough, some concerns remain to be addressed.

Major comments-

1. Previously, the authors have reported that lysosomal dysfunction and impaired autophagy contributed to pathology in Cln7 Δ ex2 mice (Brandenstein et al., 2016). Moreover, they have shown that CLN7/MFSD8 gene mutations cause depletion of soluble lysosomal proteins and impair mTOR reactivation (Danyukova T. et al., 2018). In the present study, they demonstrate that in this mouse model neurons manifest elevated mitochondrial reactive oxygen species (mROS), which leads to increased glycolysis contributing to CLN7 pathogenesis. Emerging evidence indicates that dysregulation of ER-homeostasis leads to the accumulation of misfolded proteins in the ER, which mediates the activation of unfolded protein response (UPR) (Morotta D. et al. BBA Mol Basis of Dis. 2017). Moreover, UPR in neurons may cause apoptosis. Recent reports also indicate that there is crosstalk between ER-stress and oxidative-stress, which leads to pathogenesis (Dandekar A. et al. Methods Mol Biol 1292(2015)). While the authors may have uncovered a critical piece of evidence that mitochondrial ROS induce elevated glycolysis, it would have strengthened their model if they could demonstrate that the Cln7 Δ ex2 neurons also suffer from ER-stress. A clear mechanism underlying CLN7-disease may have been uncovered if the authors could demonstrate that: ER-stress \diamond UPR \diamond mitochondrial oxidative-stress \diamond increased glycolysis \diamond Caspase-activation \diamond neuronal apoptosis \diamond CLN7 disease. While this could be a future study, the authors should at least provide some data to show whether the Cln7 Δ ex2 neurons suffer from ER-stress.

2. The authors contend that "failure of the autophagy-lysosomal pathway" causes lysosomal accumulation of mitochondria, which are structurally and functionally impaired in CLN7 disease. They also seem to suggest that ATP synthase subunit c accumulates in all NCLs. However, ATP synthase-subunit c (SCMAS) does not accumulate in all NCLs (see Tyynelä J. et al. Acta Neuropathol. 1995;89(5):391-398) although there is impairment of "autophagy-lysosomal" pathway. In Supplementary Fig.1a, the authors used SCMAS / LAMP1 ratio to demonstrate colocalization. However, the use of Pearson correlation coefficient or Mander's colocalization coefficient may have been a better way to quantify the degree of colocalization.

3. Since impaired autophagy is a characteristic finding in virtually all neurodegenerative LSDs (Seranova E. et al. Essays Biochem. 2017; Settembre C. et al. Autophagy, 2008), it is not surprising that in CLN7 disease model autophagy is also impaired. These references should be cited in the discussion.

4. In virtually all LSDs, the function of the endolysosomal system is dysregulated and recent reports indicate that the majority of the endolysosomes are in contact with the ER (Lim, C-Y. et al. Nat Cell Biol. 2019). In addition, it has been reported that the membranes for autophagosomal biogenesis are supplied by mitochondria (Haley, D.W. et al. Cell 141, 2010). This may be one of the reasons why autophagy is dysregulated by ROS-induced excessive glycolysis, which adversely affects mitochondrial survival in CLN7 disease. The authors may wish to elaborate on this in the

discussion section and suggest a possible mechanism of impaired "lysosomal-autophagy". By the way, what do the authors mean by "lysosomal-autophagy"? Functional autophagy requires the fusion of autophagosomes with lysosomes generating a hybrid organelle called autolysosome in which the cargo is degraded by lysosomal hydrolases. There are only 3 types of autophagy: macroautophagy, microautophagy and chaperone-mediated autophagy. In several LSDs the fusion of autophagosomes with lysosomes is impaired for varying reasons. Do the authors imply that autolysosome formation is impaired in Cln7 Δ ex2 neurons?

5. The damaged mitochondria in mammals is eliminated by a pathway comprising of PTEN-induced putative protein kinase 1 (PINK1) and the E3 ubiquitin ligase Parkin. The accumulation of PINK1 and Parkin on damaged mitochondria, facilitates their segregation from the mitochondrial network. This is followed by targeting of these organelles for degradation by autophagy. The data in Supplementary Fig. 1d showing elevated "PINK1 63/53 ratio" in Cln7 Δ ex2 neurons are not very convincing.

6. In the Results section, "Failure in the autophagy-lysosomal pathway.....", the data showing the levels of LC3-II or p62 are not shown. These data are needed to demonstrate the abnormality in autophagy as the authors state that the "autophagy-lysosomal pathway" is dysregulated in Cln7 Δ ex2 neurons? Demonstration of impaired autophagy flux in Cln7 Δ ex2 neurons would have been helpful in demonstrating that the impairment of the autophagy-lysosomal pathway underlies neuropathology in these mice.

7. Representative Western blots showing PFKFB3-protein (Fig. 3d) in cerebellum used β -actin as the loading control. The use of a mitochondrial protein would be a more appropriate loading control? Also, it appears that the level of RFKFB3 is lower in 2-month-old mice compared with that of the 8-month old mice. Since the level of microglia and astrocytes increase in most NCLs, including Cln7 Δ ex2 neurons, is it possible that some of the PFKFB3 originates from those cells rather than from neurons?

8. The authors suggest that targeting RFKFB3 may alleviate CLN7 pathogenesis. While this suggestion may be valid, it has been reported that for the generation of ATP, some cells like the endothelial cells (ECs), rely on glycolysis rather than oxidative phosphorylation. It has been also shown that the inhibition or loss of the glycolytic activator PFKFB3 in ECs impairs blood vessel formation. Moreover, RFKFB3 has been shown to promote "vessel sprouting" in ECs within the blood vessels and glycolysis regulates vessel branching (De Bock, K. et al. Cell 154, 2013), which is essential for the formation of new blood vessels or to repair blood vessels in the brain. Thus, inhibition of RFKFB3 may have deleterious effects if used to treat CLN7 disease. The authors should discuss this issue and state how this problem could be circumvented.

9. The authors have used both cultured Cln7 Δ ex2 neurons as well as brain tissues from Cln7 Δ ex2 mice. Since cultured neurons may not behave the same as they do in vivo, the authors should state how they have circumvented this problem in their study.

10. How was the purity (homogeneity) of the neuron cultures determined?

11. For consistency, the authors should indicate the specific location from which the cortical tissues were collected for various assays. Since neurons and other cell types in various regions of the cortex may vary greatly, it is important to mention the exact location so that the results are replicable. This could be done by providing a simple sketch of the mouse brain indicating the region of the cortex used.

Minor comments-

1. Western blot – lower case "w" is consistently used throughout the manuscript. "W" should be upper case.

2. In the magnified insets of the electron micrographs please identify the organelles like the mitochondria (M), lysosome (L) etc.

Reviewer #1 (Remarks to the Author):

Reviewer: The present manuscript examines a mouse model of neuronal ceroid lipofuscinosis. Although mitochondria are not my primary expertise, it seems to me that the evidence for mitochondrial dysfunction is good quality. The studies around glycolysis and the pentose phosphate pathway are intriguing but not fully persuasive: Changes in glycolytic flux are small, in some cases negligible (Even if nominally statistical fully significant), change in PFKFB3 levels are also small, making it hard to evaluate if the proposed main effects, much less mechanism, are valid. On the flipside, the pharmacological therapy is an impressive effort and the assays for measuring metabolism, while not very comprehensive, are generally well-chosen (Except for use of non-physiological conditions for the key radioactive assays) but not in vivo.

Authors: We are grateful for the Reviewer's positive comments about the quality of the mitochondrial dysfunction and metabolic studies. Following his/her suggestions, we have performed new experiments, including *in vivo* metabolic studies, in order to better support the main message of this work. The revised version of the manuscript contains highlighted in red color the text where the changes have been made.

Reviewer: I would suggest the following:

1. Given the main claim in the title, it seems that the Up regulation of glycolysis must be proven without a doubt. To this end, the authors need to address 4 deficiencies: a) provide in vivo evidence for glycolysis upregulation. This is admittedly not easy, but would be very desirable. Consider PET or 13C or 2H glucose tracing.

Authors: We acknowledge the Reviewer' suggestion. Indeed, we agree that this desirably *in vivo* approach is challenging, in part because we do not have the necessary facility to perform this experiment at our Institution. However, given the importance given by the Reviewer to this issue, in spite of the aggravated difficulty of transporting alive animals in a timely manner under the pandemic, we managed to involve a collaborator from a Magnetic Resonance Imaging Laboratory at CICbiomaGUNE (San Sebastian, Spain) who, having long experience in these *in vivo* analyses, accepted to perform PET and MRS studies in these animals in a relatively short time. We have included these data in the main text (Fig. 3d and Supplementary Figures 3a and 3b; text in page 7, lines 169-177, Methods in pages 32-33**).**

Unfortunately -albeit expectedly as explained below-, the 2-deoxyglucose-¹⁸F]fluoro-D-glucose ([¹⁸F]FDG) positron emission tomography (PET) *in vivo* experiment did not reveal an increase in [¹⁸F]FDG uptake in the Cln7^{Δex2} mice in any of the brain areas analyzed (**Supplementary Fig. 3a**). In part, we expected this result for two main reasons. First, as the Reviewer is aware, PET is indeed a very robust approach to detect changes in glucose uptake by brain areas; however, this technique has not the required cellular resolution to distinguish whether changes in [¹⁸F]FDG signal are due to neurons or, e.g., astrocytes glucose uptake. In our particular situation, this lack of resolution is critical, because our results obtained in Cln7^{Δex2} primary cells indicate that the change in glycolysis occurs in neurons. Given that the contribution of neurons to brain glucose uptake and glycolysis is very low, in particular when compared with astrocytes (which are highly glycolytic) (Herrero-Mendez et al., Nat Cell Biol 2009, 11:747-752; doi: 10.1038/ncb1881; Bolaños, J.

Neurochem. 2016, 139:115-125; doi: 10.1111/jnc.13486), the relatively small (although significant and reproducible) increase in glycolysis observed in Cln7^{Δex2} neurons would very hardly detectably alter the [¹⁸F]FDG signal in the brain tissue, which is mainly astrocytic (Zimmet et al., Nat Neurosci. 2017, 20:393-395; doi: 10.1038/nn.4492). Our findings indicate that the loss of Cln7 causes the stabilization of the pro-glycolytic PFKFB3 protein in neurons -not in astrocytes; however, to confirm this, we have now isolated, using an immunomagnetic approach, astrocytes from the adult brain tissue of the Cln7^{Δex2} mice, and PFKFB3 protein analyzed by Western Blotting. As shown in **Supplementary Fig. 3d**, PFKFB3 protein was unaltered in astrocytes of the Cln7^{Δex2} mice. These data further support the notion that, under our conditions, *in vivo* PET analysis will not detect changes of the [¹⁸F]FDG signal *in vivo*. Secondly, we kindly would like the Reviewer to note that we show that the increased rate in glycolysis observed in Cln7^{Δex2} neurons is associated with a concomitant decrease in the rate of glucose-6-phosphate flux through the pentose-phosphate pathway (PPP) (**Fig. 3c**). This suggests that the increased rate of glycolysis takes place at the expense of PPP, hence not necessarily affecting the overall glucose uptake of the cell -that is, the parameter measured by the [¹⁸F]FDG PET signal, as the Reviewer is aware. Nevertheless, we were not fully satisfied by not providing any piece of evidence for increased glycolysis *in vivo*. We acknowledge the ¹³C or ²H glucose tracing approaches suggested by the Reviewer. However, the analysis of the metabolites (by NMR or scintillation, respectively), which requires brain tissue homogenization, will unlikely be able to address a *specific* increase in *neuronal* glycolysis. At most, the ¹³C NMR analysis might be able to detect changes in *astrocytic* metabolism (Barros et al. Glia 2018, 66:1138-1159; doi: 10.1002/glia.23248), which is not occurring in the Cln7^{Δex2} brain. Accordingly, we decided to use a more robust metabolite analysis approach such as the ¹H-Magnetic Resonance Spectroscopy (¹H-MRS). Using the ¹H-MRS approach, we detected, in the Cln7^{Δex2} brain, a highly significant 2-fold increase in the concentration of glycine (Gly), the biosynthesis of which -along with that of serine- is strictly dependent on glycolysis *via* the phosphorylated pathway (Murtas et al., Cell Mol Life Sci 2020, 77:5131-5148; doi: 10.1007/s00018-020-03574-z; Maugard et al., Prog. Neurobiol. 2021; doi: 10.1016/j.pneurobio.2020.101896) (**Fig. 3d**). Changes in glucose or lactate were not observed (**Fig. 3d**), a result which was partly expected using this approach given that (i) as above indicated, glucose concentration is not necessarily changed under our circumstances in which there is an intracellular shift in the fluxes (decreased PPP and increased glycolysis), and (ii) lactate is either immediately released from the cells or consumed oxidatively in the form of pyruvate. We also observed changes in the concentration of certain amino acids, such as Gln, GABA or Tau, in the Cln7^{Δex2} mouse brain (**Fig. 3d**). Whilst these changes are weaker, they might also be indirectly related to the increased neuronal glycolysis –a result that is already supported *in vivo* by the observed increase in Gly- (since these amino acids carbon skeletons are originated from glucose through the tricarboxylic acid cycle), they more likely reflect alterations in neuronal neurotransmitter function(s) compatible with the neurological problems of CLN7 disease. We therefore thank the Reviewer for proposing these new experiments, which we believe have enormously strengthened the message of our work.

b) Strengthen the *in vitro* evidence by (i) Providing some insight into why the effect sizes so much smaller in 3b than 3a. (ii) measure also glucose depletion from the medium over time. This can be done with a standard glucometer, but does require careful choice of medium glucose so that the decrease is observable.

Authors:

(i) We acknowledge the Reviewer's comment for picking up this difference. However, please note that neurons of the *Cln7^{Δex2}* allele in Fig. 3a harbors the *mCAT^{LoxP}* allele too, which, even being *mCAT* inactive (in the *Cln7^{Δex2}-mCAT^{LoxP}* genotype), it is not genetically identical to the wild type neurons used for the former Fig. 3b.

Nevertheless, we admit that former Fig. 3b does not provide novel information compared with that already contained in Fig. 3a. Given this, and in order to avoid confusions but keeping identical message, we have removed the former panel 3b from this figure.

(ii) We thank the Reviewer for this suggestion. We kindly would like the Reviewer to note, as indicated in the previous point, that the net increase glucose metabolized by glycolysis in *Cln7^{Δex2}* neurons is compensated by the net decrease in glucose

metabolized by PPP (Figs. 3a,c). In good agreement with this notion, the measurements of extracellular glucose concentrations, as the Reviewer kindly suggested, did not reveal changes in *Cln7^{Δex2}* neurons *versus* wild type (please, see the figure at the right). Given that the same message is provided *in vivo* with the PET analysis, and since the space available in our manuscript is very limited, we would prefer to show these data only below for your assessment, and not include them in the revised version of the manuscript.

c) Fix deficiency of radioactive experiments by repeating at least a subset of them in some version of a normal medium not PBS (which should not be required to make radioactive measurements)

Authors: We thank the reviewer for this comment. However, we kindly would like the Reviewer to note that the radioactive experiments were not performed in PBS - we apologize if this point was unclear. In fact, the radioactive experiments were performed in a Krebs-Ringer Bicarbonate Buffer (KRPB) (NaCl 145 mM; Na₂HPO₄ 5.7 mM; KCl 4.86 mM; CaCl₂ 0.54 mM; MgSO₄ 1.22 mM; pH 7.35; 5 mM D-glucose) (page 21, lines 501-502) (Krebs, H. A. and Henseleit, K. 1932 Untersuchungen über die Harnstoffbildung im Tierkörper. Hoppe-Seyler's Zeitschrift für Physiol. Chemie 210:33-66). This medium has been widely used for biochemical fluxes experiments using radioactive tracers (e.g., please see Vicente-Gutierrez et al. 2019, 1:201-211; doi: 10.1038/s42255-018-0031-6 and the references cited therein). Nevertheless, in

order to ensure this end, following the Reviewer' suggestion, we have performed a new experiment to validate the [3-³H]-glucose conversion to ³H₂O approach to assess glycolysis in primary neurons under either KRPG or culture medium (Neurobasal, NB, medium). The rate of glycolysis was virtually identical using both incubation media (please, see the figure at the right). Given this result, and the limited space, we would like to avoid adding them to the revised manuscript and show these data only below for your assessment.

d) For oxPPP measurements, the radioactive as is really the only reliable and also easy one, but authors need to repeat in normal medium and show separately 1-¹⁴C and 6-¹⁴C data in the supplement.

Authors: We also thank the Reviewer for suggesting this experiment and requesting for the separate 1-¹⁴C and 6-¹⁴C data. Regarding the medium, we would like to clarify, as for the previous, point, that the PPP measurements were performed in KRPG -not PBS- medium, a condition that has been previously validated (e.g., please see Vicente-Gutierrez et al. 2019, 1:201-211; doi: 10.1038/s42255-018-0031-6;

Herrero-Mendez et al., Nat Cell Biol 2009, 11:747-752; doi: 10.1038/ncb1881). Nevertheless, following the Reviewer' suggestion, we performed an experiment to validate this end, which revealed that ¹⁴CO₂ production was identical in neurons incubated under KRPG or Neurobasal (NB) medium (please, see the figure at the left). We fully agree with the Reviewer that 1-¹⁴C and 6-¹⁴C data should have been shown, and we apologize for not having done so in the first version. We have now included this information in the new version (Supplementary Dataset, sheet for Figure 3c). As therein shown, the decrease in ¹⁴CO₂ formation from [1-¹⁴C]glucose was ~50 nmol/h x mg protein, whereas the decrease in ¹⁴CO₂ formation from [6-¹⁴C]glucose was ~18 nmol/h x mg protein.

2) I would suggest changing "mediates" to "in" in the title, so the Authors have less pressure to prove that a major damage pathway goes through glycolysis, for which a single pharmacological manipulation is insufficient.

Authors: We thank the reviewer for this suggestion and therefore we have amended the title as suggested.

3) Put some of the key data in the supplement on lactate secretion and PFKFB3 quantitation into the maintext. The PFKFB3 blots are not persuasive but the quantitation shows a trend that is statistically significant across samples. Also, weaken claims by explicitly pointing out that these are small effect size in abstract and main text.

Authors: OK. Following the Reviewer' suggestion, we moved some of the lactate release and PFKFB3 quantitation into the main text (Fig. 3b,f). Regarding the effect size, we agree that this is not very persuasive in the PFKFB3 blots as stands, in spite the increase is consistent across samples reaching the statistical significance. However, we would like the Reviewer to note that, for the *in vivo* PFKFB3 measurements, we used brain tissue hence containing a mixture of different cell types besides neurons. This fact may underestimate the size of effects. Therefore, we have now acutely isolated neurons from the adult (5-months-old) *Cln7^{Δex2}* and wild type brain tissue using an immunomagnetic approach, and neuronal extracts were used for immunoblotting. As shown in the new **Supplementary Fig. 3d, immunomagnetically separated neurons (β -tubulin-III-positive) from the *Cln7^{Δex2}* mice revealed an increase in PFKFB3 protein levels that is much greater in size than those obtained from whole tissue (**page 8, lines 186-190**). This result strongly suggests that the apparently small size in the PFKFB3 signal from brain whole-tissue extracts may be underestimated. At the light of these new data, we kindly request to avoid adding adjectives to specify effect sizes, as long as they are statistically significant. To describe the size of this and other effects we have indicated fold changes the first time they are mentioned in the text (**page 7, lines 164 and 181; page 8, line 185**).**

4) Change "pharmacological inhibition of PFKFB3" in abstract to the name of the agent as a single pharmacological agent is not sufficient to prove that the activity flows through the protein target without genetics, *in vivo* target engagement validation, or rescue, all of which are missing (or provide 1 of these).

Authors: We agree with the Reviewer with his/her comment. We opted for the Reviewer' suggestion to indicate the name of the agent in the abstract.

5) Add "may" to 1st part of the conclusion sentence of abstract. "Aberrant upregulation of neuronal glycolysis MAY contribute to CLN7 pathogenesis"

Authors: We thank the Reviewer for his/her comment on the concluding sentence, which we addressed. To avoid two "may" in the same sentence, we changed the second "may" to "could".

6) Better explain methods for F26BP measurement, provide raw data supporting the measurement in the supplement, note that effect size here also is small.

Authors: We thank the Reviewer for these comments. We have now explained F26BP determination in methods (page 22, lines 534-539**). The raw data are provided in the **Supplementary Dataset document (sheet for Fig. 3e)**. We have specified the fold change increase in the text (**page 7, line 181**).**

Reviewer #2 (Remarks to the Author):

Reviewer: In this study, Lopez-Fabuel et al. investigated that abnormal upregulation of glycolysis, which is due to PFKFB3 stabilization, contributes to CLN7 neuronal ceroid lipofuscinosis (NCLs).

I recommend the work can be published but the following queries need to be addressed.

Authors: We thank the Reviewer for his/her useful and constructive comments. The revised version of the manuscript contains highlighted in red color the text where the changes have been made.

1. In figure 3l, p35 western blot data is not accordant with supplementary figure 3i. The p35 p25 protein expression level in both Cln7 Δ ex2 groups (with or without MDL) looks like similar.

Authors: We apologize for the apparent confusion generated by the way these data were shown in the original version. We have now repeated these experiments, the Western blots were now loaded in the correct order, and the band intensities of the replicas were quantified. The results are now shown in **Fig. 3m and Supplementary Fig. 3i. We hope that the new data are now convincing.**

2. The authors need to explain why Cdk5 protein expression is upregulated in Cln7 Δ ex2 neurons. Have that phenomenon been well known in CLN7 neuronal ceroid lipofuscinosis? The increased Cdk5 protein expression may also affect hyperphosphorylation of Cdh1.

Authors: We thank the reviewer for this comment. We would like to kindly ask the Reviewer to note that we have not found Cdk5 protein upregulation in Cln7 Δ ex2. As shown in **Fig. 3l,m (formerly 3k, l), we observed an increased conversion of p35 to p25, a proteolytic activity that is catalyzed by calpain (please, see **Fig. 3m** depicting the rescue of p35 conversion to p25 by the calpain inhibitor MDL). Since calpain activity is known to be activated by Ca²⁺, which we found elevated (**Fig. 3i, formerly 3h**), and Ca²⁺ quelation (with BAPTA), prevented PFKFB3 stabilization (**Fig. 3j, formerly 3i**), we inferred that calpain-mediated p35 conversion to p25 would be a step of the molecular cascade. Indeed, p25 is a potent Cdk5 activator, but it does not affect Cdk5 protein expression. As the Reviewer correctly states, active Cdk5 causes Cdh1 hyperphosphorylation (Maestre et al., 2008, EMBO J. 27:2736-2745; doi: 10.1038/emboj.2008.195), a situation that we herein show in the Cln7 Δ ex2 neurons (**Fig. 3h, formerly 3g**). We thank the Reviewer for his/her interesting query on the possible link between Cdk5 and neuronal ceroid lipofuscinosis. Whilst at the time of writing the manuscript we found no literature linking Cdk5 and CLN7 disease, following the Reviewer' suggestion, we made a deeper literature search and found a recent article relating Cdk5 with CLN6 neuronal ceroid lipofuscinosis. Thus, the compound (S)-lacosamide (S-N-benzy-2-acetamido-3-methoxypropionamide), a Cdk5 inhibitor, prevents the phosphorylation of the collapsin response mediator protein 2 (CRMP2) (Moutal et al., 2016, Mol. Neurobiol. 53:1959-1976; doi.org/10.1007/s12035-015-9141-2), a cytoskeletal protein that becomes reduced when is hyperphosphorylated. Interestingly, CRMP2 has been found reduced in CLN6^{ncl^f} mutant mouse model of neuronal ceroid lipofuscinosis (Benedict et al. 2009,**

J. Neurosci. Res. 87:2157-2166; doi.org/10.1002/jnr.22032; Ip et al., 2014, Neuroscientist 20:589-598; doi.org/10.1177/1073858413514278). Moreover, incubating Cln6^{ncif} mouse neurons and CLN6 disease patient fibroblasts with (S)-lacosamide showed restoration of lysosomal associated deficits, although unfortunately it did not exert protection on the behavioral and survival outcomes of the disease *in vivo* (White KA et al., 2019, Neuronal Signaling 3:NS20190001; doi.org/10.1042/NS20190001). Despite being ineffective *in vivo* in the Cln6^{ncif} mouse model, these results are indeed very relevant because they share with our work a single molecular step (Cdk5 activity) that belongs to the molecular cascade that leads to PFKFB3 protein stabilization (in the Cln7^{Δex2} mouse). Whether (S)-lacosamide is effective *in vivo* in the Cln7^{Δex2} mouse model is unknown; however, given that PFKFB3 inhibition (by AZ67) was effective in the Cln7^{Δex2} mouse (our data), it would be interesting to test in the future whether AZ67 is effective also in the CLN6 disease. We have discussed this issue in the amended version of the manuscript (**page 10, lines 256 and page 11, lines 257-259**).

3. Please insert 'h' in the figure 3. Figure 3h is missing.

Authors: OK. Thanks for noting. We amended accordingly.

4. The authors should explain what Pa474 is in Figure 4h in the manuscript.

Authors: We thank the Reviewer for this comment. We have now further explained the patients' descriptions for these experiments (**page 15, lines 375-378**).

5. The manuscript text should be carefully edited. There are errors such as '...such as CLN2/TPP1 (Ref. 11), here ...' (p3, line 71), and '...a robust positive effector of PFK1 (Ref. 26). The rate...' (p7, line 158).

Authors: OK. Thanks for noting these errors, which we now amended accordingly.

6. Please mention RRID of antibodies in the manuscript.

Authors: OK. Thanks for noting this, which we have now fixed accordingly (please, see **Supplementary Information document (page 6, lines 138-140)**).

Reviewer #4 (Remarks to the Author):

General comments-

Using Cln7 Δ ex2 mice, a reliable animal model of CLN7-disease, Lopez-Fabuel and colleagues have conducted a comprehensive study to explore the mechanism of pathogenesis underlying CLN7 disease. Their results are impressive and clearly demonstrate that mitochondrial oxidative stress causing increased glycolysis, at least in part, contributes to CLN7 pathogenesis. They further demonstrate that the use of pharmacological agents targeting RFKFB3 may be a potential therapeutic target for this disease. While their investigations are thorough, some concerns remain to be addressed.

Authors: We thank the Reviewer for his/her constructive and thoughtful comments on our manuscript. The revised version of the manuscript contains highlighted in red color the text where the changes have been made.

Major comments-

1. Previously, the authors have reported that lysosomal dysfunction and impaired autophagy contributed to pathology in Cln7 Δ ex2 mice (Brandenstein et al., 2016). Moreover, they have shown that CLN7/MFSD8 gene mutations cause depletion of soluble lysosomal proteins and impair mTOR reactivation (Danyukova T. et al., 2018). In the present study, they demonstrate that in this mouse model neurons manifest elevated mitochondrial reactive oxygen species (mROS), which leads to increased glycolysis contributing to CLN7 pathogenesis. Emerging evidence indicates that dysregulation of ER-homeostasis leads to the accumulation of misfolded proteins in the ER, which mediates the activation of unfolded protein response (UPR) (Morotta D. et al. BBA Mol Basis of Dis. 2017). Moreover, UPR in neurons may cause apoptosis. Recent reports also indicate that there is crosstalk between ER-stress and oxidative-stress, which leads to pathogenesis (Dandekar A. et al. Methods Mol Biol 1292(2015). While the authors may have uncovered a critical piece of evidence that mitochondrial ROS induce elevated glycolysis, it would have strengthened their model if they could demonstrate that the Cln7 Δ ex2 neurons also suffer from ER-stress. A clear mechanism underlying CLN7-disease may have been uncovered if the authors could demonstrate that: ER-stress UPR mitochondrial oxidative-stress increased glycolysis Caspase-activation neuronal apoptosis CLN7 disease. While this could be a future study, the authors should at least provide some data to show whether the Cln7 Δ ex2 neurons suffer from ER-stress.

Authors: We thank the Reviewer for suggesting such an interesting molecular cascade to link ER-stress with ROS and glycolysis in CLN7. We were highly enthusiastic about this possibility and, although as the Reviewer states, this could be suitable for a future study, we decided to test for ER-stress in Cln7 Δ ex2 neurons. Accordingly, we performed RT-qPCR to assess the RNA abundances of key UPR proteins, namely total and spliced X-box binding protein 1 (XBP1 and sXBP1), activating transcription factor 4 (ATF4), C/EBP homologous protein (CHOP), binding immunoglobulin protein (BiP), and endoplasmic reticulum degradation-enhancing alpha-mannosidase-like protein 1 (EDEM1), which were normalized against β -actin RNA abundance. Unfortunately, we found no differences between Cln7 Δ ex2 and wild

type neurons (**Supplementary Fig. 1i**). It should be mentioned that, in cerebellar extracts from the CLN6 LSD mouse $Cln6^{cnlf}$ model, all markers of UPR analyzed were found unchanged by other authors (Thelen et al. 2012, PLOS One 7: e35493; doi: 10.1371/journal.pone.0035493). Although there are potential differences in the molecular cascade of events between CLN6 and CLN7 diseases, collectively, these data are not supportive for a potential involvement of ER-stress in this pathology, at least under our experimental conditions. Nevertheless, we thank the Reviewer for this suggestion, which will be further pursued in the future. Given the interest of this pathway, in spite of being a negative result, we have included them in the amended version of the manuscript (**pages 4, lines 106-107; page 5, lines 108-110**).

2. The authors contend that “failure of the autophagy-lysosomal pathway” causes lysosomal accumulation of mitochondria, which are structurally and functionally impaired in CLN7 disease. They also seem to suggest that ATP synthase subunit c accumulates in all NCLs. However, ATP synthase-subunit c (SCMAS) does not accumulate in all NCLs (see Tyynelä J. et al. Acta Neuropathol. 1995;89(5):391-398) although there is impairment of “autophagy-lysosomal” pathway. In Supplementary Fig.1a, the authors used SCMAS / LAMP1 ratio to demonstrate colocalization. However, the use of Pearson correlation coefficient or Mander’s colocalization coefficient may have been a better way to quantify the degree of colocalization.

Authors: We acknowledge the Reviewer for the reference showing lack of SCMAS accumulation in all NCLs (Tyynelä J. et al. Acta Neuropathol. 1995;89(5):391-398), which we have now cited (**page 6, line 147-148**) to clarify this point. Regarding the **Supplementary Fig. 1b**, we indeed used the Mander’s colocalization coefficient to quantify the degree of colocalization; we apologize for not having indicated this in the figure, although it was mentioned in Methods. We have now amended the figure (**Supplementary Fig. 1b**).

3. Since impaired autophagy is a characteristic finding in virtually all neurodegenerative LSDs (Seranova E. et al. Essays Biochem. 2017; Settembre C. et al. Autophagy, 2008), it is not surprising that in CLN7 disease model autophagy is also impaired. These references should be cited in the discussion.

Authors: We fully agree with the Reviewer and acknowledge the suggestion to cite these references. They have now been cited (**references 13 and 14, cited in page 4, lines 83-84**).

4. In virtually all LSDs, the function of the endolysosomal system is dysregulated and recent reports indicate that the majority of the endolysosomes are in contact with the ER (Lim, C-Y. et al. Nat Cell Biol. 2019). In addition, it has been reported that the membranes for autophagosomal biogenesis are supplied by mitochondria (Haley, D.W. et al. Cell 141, 2010). This may be one of the reasons why autophagy is dysregulated by ROS-induced excessive glycolysis, which adversely affects mitochondrial survival in CLN7 disease. The authors may wish to elaborate on this in the discussion section and suggest a possible mechanism of impaired “lysosomal- autophagy”. By the way, what do the authors mean by “lysosomal-autophagy”? Functional autophagy requires the fusion of autophagosomes with lysosomes generating a hybrid organelle called autolysosome in which the cargo is degraded by lysosomal hydrolases. There are only 3

types of autophagy: macroautophagy, microautophagy and chaperone-mediated autophagy. In several LSDs the fusion of autophagosomes with lysosomes is impaired for varying reasons. Do the authors imply that autolysosome formation is impaired in Cln7 Δ ex2 neurons?

Authors: We acknowledge the Reviewer for his/her comment on the different mechanisms whereby the fusion of autophagosomes with lysosomes may be impaired in LSDs and, in particular, in the Cln7 Δ ex2 model. We admit we have not specifically designed experiments to address this question. According to our new data shown in **Fig. 1b** (suggested by the Reviewer under the point #6, below) we can only assert that there is an impairment in macroautophagy in Cln7 Δ ex2 neurons. Whether this impairment occurs by autophagosome accumulation secondary to loss of fusion with lysosomes, by autolysosome accumulation, or both, is currently unknown. Whilst this specific issue is not the main focus of our manuscript, we admit that it would be a very important question to solve, particularly given that the actual function of CLN7 is unknown. Addressing this issue would require a substantial amount of experiment that exceeds the scope of this study. However, we acknowledge the Reviewer's suggestion to elaborate a possible mechanism in the Discussion section based on the possibility that, given that the membranes for autophagosomal biogenesis might be supplied by mitochondria (Haley, D.W. et al. Cell 141, 2010), the observed impaired autophagy may be consequence of a loss of mitochondrial support of autophagosomal biogenesis, a possibility that has now been discussed (please, see **page 11, lines 274-277**). We thank the Reviewer for this interesting suggestion.

5. The damaged mitochondria in mammals is eliminated by a pathway comprising of PTEN-induced putative protein kinase 1 (PINK1) and the E3 ubiquitin ligase Parkin. The accumulation of PINK1 and Parkin on damaged mitochondria, facilitates their segregation from the mitochondrial network. This is followed by targeting of these organelles for degradation by autophagy. The data in Supplementary Fig. 1d showing elevated "PINK1 63/53 ratio" in Cln7 Δ ex2 neurons are not very convincing.

Authors: We thank the Reviewer for his/her comment. We are aware of the weak effect on PINK1 63/53 ratio. Accordingly, we have now performed a new experiment to assess Parkin in isolated mitochondria from Cln7 Δ ex2 neurons. As shown in the new **Supplementary Fig. 1e** (see, also, **page 4, line 92**), we found Parkin accumulation in Cln7 Δ ex2 mitochondria too, which strengthen the notion of an impaired damaged mitochondrial clearance in Cln7 Δ ex2 neurons.

6. In the Results section, "Failure in the autophagy-lysosomal pathway.....", the data showing the levels of LC3-II or p62 are not shown. These data are needed to demonstrate the abnormality in autophagy as the authors state that the "autophagy-lysosomal pathway" is dysregulated in Cln7 Δ ex2 neurons? Demonstration of impaired autophagy flux in Cln7 Δ ex2 neurons would have been helpful in demonstrating that the impairment of the autophagy-lysosomal pathway underlies neuropathology in these mice.

Authors: We fully agree with the Reviewer's comment and we acknowledge for the suggested experiment. Accordingly, we have performed a new experiment in which wild type or Cln7 Δ ex2 neurons were incubated with lysosomal inhibitors (leupeptin

and ammonium chloride), and the levels of LC3-II assessed by Western blotting. As shown in **Fig. 1b** (see, also, page 3, lines 81-82 and page 4, lines 83-84), we found autophagic flux in the wild type neurons; in addition, we observed that $Cln7^{\Delta ex2}$ neurons tend to accumulate LC3-II that was not further enhanced by lysosomal inhibition, indicating impaired autophagy.

7. Representative Western blots showing PFKFB3-protein (Fig. 3d) in cerebellum used β -actin as the loading control. The use of a mitochondrial protein would be a more appropriate loading control? Also, it appears that the level of PFKFB3 is lower in 2-month-old mice compared with that of the 8-month old mice. Since the level of microglia and astrocytes increase in most NCLs, including $Cln7^{\Delta ex2}$ neurons, is it possible that some of the PFKFB3 originates from those cells rather than from neurons?

Authors: We believe that, being PFKFB3 a cytosolic protein, the use of β -actin would be a better loading control than a mitochondrial protein. In addition, given that mitochondria are aberrantly accumulated in the $Cln7^{\Delta ex2}$ brain, the use of a mitochondrial protein as loading control may complicate the interpretation of the results. Therefore, we respectfully ask to keep β -actin as the loading control. Regarding the apparently lower levels of PFKFB3 in 2 months-old mice compared with that of the 8 months-old mice, we kindly ask the Reviewer to note that the Western blot shown in **Supplementary Fig. 3c** (formerly S3d) is one of the three replica Western blots, which were performed to obtain a statistically robust quantification of the band intensities. As shown in the document “Suppl replicas 2”, there are two more Western blots, which were used along with that shown in **Supplementary Fig. 3c** (formerly S3d) to obtain the graph depicted by the right-hand side of the Western blot in **Supplementary Fig. 3c** (formerly S3d), which reveals an overall higher increase in PFKFB3 abundance in the $Cln7^{\Delta ex2}$ cerebellum at 8 months versus 2 months. The Reviewer’s comment on the possible involvement of increased glial reactivity in CLN7 disease as a possible contributing factor in PFKFB3 signal is very interesting. Accordingly, we have undertaken new experiments specifically focused to address this issue. Thus, we have used an immunomagnetic approach to separate neurons from glial cells in adult (5-months-old) $Cln7^{\Delta ex2}$ and wild type cerebella. As shown in the new **Supplementary Fig. 3d**, immunomagnetically separated cells (i.e., neurons; β -tubulin-III-positive) from the $Cln7^{\Delta ex2}$ cerebella showed a ~4.8-fold increase in PFKFB3 protein levels. However, immunomagnetically separated glial cells (i.e., a mixture of astrocytes, microglia and oligodendrocytes; GFAP-positive indicating astrocytic enrichment) showed no statistically significant changes in PFKFB3 protein abundance (**Supplementary Fig. 3d**). Altogether, these new data disregard the possibility that the PFKFB3 signal comes from glial cells. We have described these data in the new version of the manuscript (**page 8, lines 186-190**).

8. The authors suggest that targeting RFKFB3 may alleviate CLN7 pathogenesis. While this suggestion may be valid, it has been reported that for the generation of ATP, some cells like the endothelial cells (ECs), rely on glycolysis rather than oxidative phosphorylation. It has been also shown that the inhibition or loss of the glycolytic activator PFKFB3 in ECs impairs blood vessel formation. Moreover, RFKFB3 has been shown to promote “vessel sprouting” in ECs within the blood vessels and glycolysis regulates vessel branching (De Bock, K. et al. *Cell* 154, 2013), which is essential for the formation of new blood vessels or to repair blood vessels in the brain. Thus, inhibition of RFKFB3 may have deleterious effects if used to treat CLN7 disease. The authors should discuss this issue and state how this problem could be circumvented.

Authors: We acknowledge the Reviewer for noting this interesting issue. Given the negative impact that the possible interference of PFKFB3 inhibition on vessel sprouting would exert during the possible future use of AZ67 in the clinic, we have undertaken specific experiments to address this possibility. We have performed two types of experiments. First, we performed an *in vitro* study on an endothelial cell line (human umbilical vein endothelial cells, HUVECs) to test the potential impairment of AZ67 on VEGF-A-induced cell sprouting, using AZ67 at doses within and well above the range of those used in Cln7^{Δex2} neurons. As shown in the **figure 1 below**, AZ67 failed to interfere with HUVEC sprouting. And second, we analyzed by immunohistochemistry the expression of the endothelial cell marker CD31 in the brain cortex of 5-months-old wild type and Cln7^{Δex2} mice, treated with AZ67 for the last 2 months i.c.v., as indicated in the Methods section. As shown in the **figure 2 below (next page)**, we found no differences in CD31 staining, suggesting that the brain microvasculature remains intact during this long-term AZ67 treatment. Altogether, these data indicate that, at least under these circumstances, PFKFB3 inhibition by AZ67 does not negatively impact on the brain vessel sprouting. We believe that these new data are very important for a possible future use of this compound; however, being a negative result, given that it does not directly affect to the main message of our work, and since the limited available space for our manuscript, we would prefer to avoid adding them to the revised version of the manuscript.

Figure 1. Cellular Angiogenesis Assay. Human umbilical vein endothelial cells (HUVEC) (PromoCell) spheroids were prepared as described (Korff & Augustin, 1998) by pipetting 500 HUVEC in a hanging drop on plastic dishes to allow overnight spheroid aggregation. Fifty HUVEC spheroids were seeded in 0.9 ml of a collagen gel and pipetted into individual wells of a 24 well plate to allow polymerization. AZ67, at the concentrations shown in the graph, in combination with the growth factor (VEGF-A; 25 ng/ml) was added after 30 min on the top of the polymerized gel. Two types of controls were used, namely vehicle with VEGF-A and vehicle

without VEGF-A. Plates were incubated at 37°C for 24 hours and fixed by adding 4% PFA. Sprouting intensity of HUVEC spheroids were quantified by an image analysis system determining the cumulative sprout length per spheroid (CSL) using an inverted microscope and the digital imaging software NIS Elements BR 3.0 (Nikon). The mean of the cumulative sprout

length of 10 randomly selected spheroids was analyzed as an individual data point. Korff T & Augustin HG (1998) Integration of endothelial cells in multicellular spheroids prevents apoptosis and induces differentiation. *J Cell Biol* 143:1341-1352. doi: 10.1083/jcb.143.5.1341.

Cortex

Figure 2. CD31 (1/100 dilution of anti-CD31; cat. Number 550274, Becton Dickinson) immunohistochemical analysis of the mouse brain cortex after 2 months of a daily intracerebroventricular administration of AZ67 (1 nmol/mouse) (3 serial slices per mouse; n=3 of 3-months-old mice). Scale bar, 100 μ m. (Please, refer to Methods section for further methodological details).

9. The authors have used both cultured *Cln7 Δ ex2* neurons as well as brain tissues from *Cln7 Δ ex2* mice. Since cultured neurons may not behave the same as they do *in vivo*, the authors should state how they have circumvented this problem in their study.

Authors: We thank the reviewer for this comment. Indeed, we think that, even being primary cells, the cultured neurons from *Cln7 Δ ex2* mice do not necessarily have to behave exactly as neurons do *in vivo*. Thus, we are aware of the influence of neighbor astrocytes (and other cell types) in the intact brain. However, we believe that the use of a set of primary cells enriched in neurons, even with the drawbacks derived from the fact that they are living in an artificial surrounding, is a highly reproducible and widely established model for the specific study of molecular mechanisms occurring intracellularly in a particular type of enriched cell population. In spite of this, we are aware of the limitations to extrapolate the findings obtained in primary neurons to the *in vivo* situation. It is precisely because of this limitation we have re-analyzed the key molecular findings and pharmacological effects that we found in primary neurons, on the brain of the *Cln7 Δ ex2* *in vivo* mouse model. With both approaches, i.e., primary neurons and *in vivo* validation on the brain, we are confident that the key message of our work is well supported by the results. Given the concern raised by the Reviewer about this issue, we have stated a note in the text of the amended version of the manuscript (page 5, lines 110-112).

10. How was the purity (homogeneity) of the neuron cultures determined?

Authors: We thank the Reviewer for this comment. Accordingly, we have performed an immunocytochemical analysis of the cell type composition of the primary neuronal preparation used in this study, which revealed a 99.02% enrichment in neurons (β -tubulin-III), 0.43% astrocytes (GFAP), 0.11% oligodendrocytes (O4), 0.13% microglia (CD45) and 0.31% other unidentified cells. We have included this information in the text (page 15, lines 375-378).

11. For consistency, the authors should indicate the specific location from which the cortical tissues were collected for various assays. Since neurons and other cell types in various regions of the cortex may vary greatly, it is important to mention the exact location so that the results are replicable. This could be done by providing a simple sketch of the mouse brain indicating the region of the cortex used.

Authors: We fully agree with the Reviewer and, accordingly, we have now included a sketch of the mouse brain indicating the regions of the brain that were used for the different types of analyses (Supplementary Fig. 1j).

Minor comments-

1. Western blot – lower case “W” is consistently used throughout the manuscript. “W” should be upper case.

Authors: Thanks for noting this, which has now been amended.

2. In the magnified insets of the electron micrographs please identify the organelles like the mitochondria (M), lysosome (L) etc.

Authors: We thank the Reviewer for this suggestion. Accordingly, we have now identified these organelles in all EM images.

Reviewers' Comments:

Reviewer #1:

Remarks to the Author:

The present revision presents critical new data, collected by experts, that assesses the title claim of "aberrant upregulation of glycolysis in CLN7 neuronal ceroid lipofuscinosis." Overall, with credit to the authors for their honest presentation of the results, this new data argues against the title claim. The statements in the text line 172-175 do not support the claim "The data indicate increased glycolytic flux in Cln7ko mouse brain in vivo." Glycine is a minor product of glycolysis and cannot be used to read out overall glycolytic flux. Accordingly, while it is possible that glycolytic flux is indeed increased in specific brain cell types (neurons) and the data in Fig S3d are impressive, the title claim is unproven and accordingly the manuscript is unacceptable. The authors may wish to consider reframing the manuscript in a manner that better ties the title to the observations, e.g. a title focused on PFKFB3 in neurons in this disease model could potentially be better supported. Alternatively, they may wish to consider modern methods for cell-type specific glycolytic measurement in vivo as exemplified by recent work in Nature by Rathmell and colleagues (2021).

Reviewer #2:

Remarks to the Author:

Thank you for your detailed comment to reviewer's remarks.
I recommend this work can be published.

Reviewer #4:

Remarks to the Author:

I reviewed the revised manuscript and the responses provided by the authors to my comments and I am satisfied that the authors have been adequately addressed all my comments and added new data to make this paper more solid.
Therefore, I have no further comments on this revised manuscript.

Reviewer #1 (Remarks to the Author):

Reviewer: The present revision presents critical new data, collected by experts, that assesses the title claim of "aberrant upregulation of glycolysis in CLN7 neuronal ceroid lipofuscinosis." Overall, with credit to the authors for their honest presentation of the results, this new data argues against the title claim. The statements in the text line 172-175 do not support the claim "The data indicate increased glycolytic flux in Cln7ko mouse brain in vivo." Glycine is a minor product of glycolysis and cannot be used to read out overall glycolytic flux. Accordingly, while it is possible that glycolytic flux is indeed increased in specific brain cell types (neurons) and the data in Fig S3d are impressive, the title claim is unproven and accordingly the manuscript is unacceptable. The authors may wish to consider reframing the manuscript in a manner that better ties the title to the observations, e.g. a title focused on PFKFB3 in neurons in this disease model could potentially be better supported. Alternatively, they may wish to consider modern methods for cell-type specific glycolytic measurement in vivo as exemplified by recent work in Nature by Rathmell and colleagues (2021).

Authors: We are grateful for the thoughtful comment by the Reviewer. Indeed, we admit that an increase in glycine concentration, as observed by [¹H]MRS, is not a canonical readout of glycolysis. We have therefore mentioned this circumstance in the text, along with a decreased tone in the claim that we were able to detect an increased glycolysis in vivo in the Cln7^{Δex2} mouse brain. As suggested by the Reviewer, we have focused on the observation that pro-glycolytic enzyme PFKFB3 is enhanced in CLN7 disease. Therefore, we changed the title, the abstract and the text accordingly, as follows:

The title now reads:

"Aberrant upregulation of pro-glycolytic enzyme PFKFB3 in CLN7 neuronal ceroid lipofuscinosis"

The abstract now reads, in its last sentence:

"Thus, aberrant upregulation of pro-glycolytic enzyme PFKFB3 in neurons may contribute to CLN7 pathogenesis and targeting PFKFB3 could alleviate this and other lysosomal storage diseases."

The text now reads (lines 171-177):

"However, in vivo ¹H-magnetic resonance spectroscopy ([¹H]MRS) analysis of the Cln7^{Δex2} mouse brain revealed a twofold increase in the concentration of glycine (Supplementary Fig. 3b,c). Whilst the biosynthesis of glycine via the phosphorylated pathway requires glycolysis³⁰, its concentration is not a direct evidence of the glycolytic flux. Therefore, using [¹⁸F]FDG-PET and [¹H]MRS, being approaches that lack cell-level resolution, failed to unambiguously ascertain in vivo up-regulation of neuronal glycolysis in Cln7^{Δex2} mice."